# Solving New Tasks by Adapting Internet Video Knowledge

**Calvin Luo**[*,1] , **Zilai Zeng**[*,1], **Yilun Du**[2], **Chen Sun**[1]
[1]Brown University, [2]Harvard University

## Abstract

Video generative models demonstrate great promise in robotics by serving as visual planners or as policy supervisors. When pretrained on internet-scale data, such video models intimately understand alignment with natural language, and can thus facilitate generalization to novel downstream behavior through text-conditioning. However, they may not be sensitive to the specificities of the particular environment the agent inhabits. On the other hand, training video models on in-domain examples of robotic behavior naturally encodes environment-specific intricacies, but the scale of available demonstrations may not be sufficient to support generalization to unseen tasks via natural language specification. In this work, we investigate different adaptation techniques that integrate in-domain information with large-scale pretrained video models, and explore the extent to which they enable novel text-conditioned generalization for robotic tasks, while also considering their independent data and resource considerations. We successfully demonstrate across robotic environments that adapting powerful video models with small scales of example data can successfully facilitate generalization to novel behaviors. In particular, we present a novel adaptation strategy, termed *Inverse Probabilistic Adaptation*, that not only consistently achieves strong generalization performance across robotic tasks and settings, but also exhibits robustness to the quality of adaptation data, successfully solving novel tasks even when only suboptimal in-domain demonstrations are available.

## 1 Introduction

Recent advancements in video generative models have made promising their application to interactive problems and decision-making. Such video models trained explicitly on in-domain demonstrations have demonstrated accurate encoding of environment-specific visual details and dynamics, and have been popular choices to utilize for robotic learning (Du et al., 2024b; Huang et al., 2023; Yang et al., 2023b; Ko et al., 2024; Liang et al., 2024). When optimized on expert video demonstrations, their encoded understanding of expert behavior can be directly used to supervise the learning of high-performing policies (Huang et al., 2023; Escontrela et al., 2024), and applied as performant visual planners (Du et al., 2024a) in robotic settings. However, for arbitrary robotic environments, there is usually a severe difference in scale of tractably available expert demonstration data, especially with associated text labelling, in comparison with general internet-scale datasets of videos paired with natural language. As a result, such in-domain video generative models usually suffer from weaker generalization capability, across novel text specifications and motions of interest.

Instead of training directly on in-domain demonstrations, stronger generalization performance can be obtained by using text-to-video models pretrained on internet-scale data. Having summarized powerful priors over visual styles, natural motion, and alignment with natural language from large-scale data, such models can be leveraged to supervise policies that behave flexibly conditioned on text across multiple environment visual styles without modification (Luo et al., 2024). However, interaction is often performed in a fixed environment with specific visual characteristics and potentially unique interaction dynamics, which a general environment-agnostic video model may not inherently understand or respect. Thus, directly applying a large-scale pretrained video generative model

---

*: Equal contribution. Correspondence to: calvin_luo@brown.edu and zilai_zeng@brown.edu.

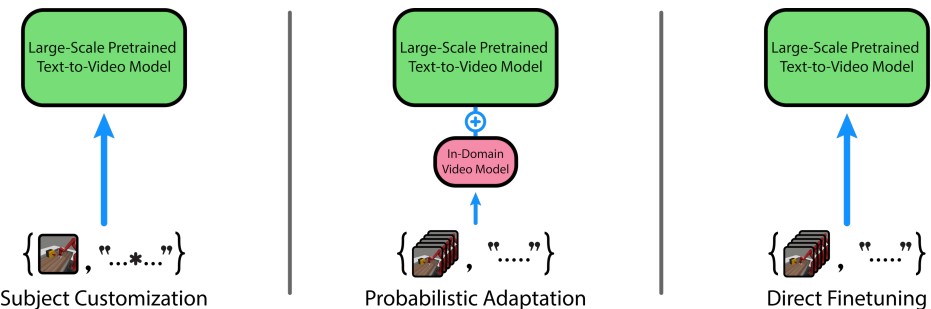

Figure 1: **Adaptation Techniques.** We explore how in-domain information can be integrated into large-scale text-to-video models through three different adaptation techniques: Subject Customization, Probabilistic Adaptation, and Direct Finetuning. Subject Customization only modifies the image and text encoder, rather than the motion module, and is lightweight in terms of data requirements: it only utilizes pairs of static images and text annotated with a special identifier. Probabilistic Adaptation learns a small in-domain model from paired video data, which is then used through score composition with a large-scale video model that is kept frozen. The small in-domain model can be flexibly parameterized to consider available training resources. Direct Finetuning seeks to update the motion module of the large-scale pretrained video model with in-domain paired video data.

without modification comes with a potential drawback; they may not understand the intricacies of particular environments of interest to successfully facilitate high-performance behavior within them.

These considerations naturally motivate the investigation of ways to mutually cover the independent deficiencies of each approach. In this work, we perform a thorough study on novel task generalization via adapting internet video knowledge; we seek to illuminate how in-domain information can be best integrated into large-scale pretrained text-to-video models, such that powerful zero-shot text-conditioned generalization capabilities are enabled while considering environment-specific knowledge pertaining to visual styles and interaction dynamics. We compare the downstream robotic performance of multiple adaptation techniques and contrast their respective requirements on in-domain data samples, which range from utilizing only a few still-frames of the agent to text-labelled video demonstrations, and training resources, which span from direct finetuning of the large-scale video model to utilizing it only for inference without any updates. Broadly, we provide this study of adaptation for facilitating action prediction (**Adapt2Act**) as valuable insight to the practitioner interested in balancing performance with resource availability.

We perform standardized evaluations across both robotic manipulation tasks (Yu et al., 2020) and continuous control (Tassa et al., 2018), and demonstrate that adapted video generative models are able to successfully act as accurate video planners for novel text-conditioned specifications across a variety of robotic tasks, and can also supervise the learning of novel text-conditioned policies. Furthermore, in this work we propose a novel adaptation technique termed *Inverse Probabilistic Adaptation*, which we highlight achieves consistently strong generalization performance across robotic environments and downstream evaluation approaches. We also discover that even when only suboptimal in-domain demonstrations are provided, it can still effectively leverage web-scale priors and text conditioning to generate coherent video plans and successfully solve novel tasks. Visualizations and code are provided at diffusion-supervision.github.io/adapt2act/.

## 2 RELATED WORK

**Adaptation Techniques for Diffusion Models.** Although many large-scale pretrained text-to-video models (Ho et al., 2022; Guo et al., 2023; Ramesh et al., 2022; Xing et al., 2023; Villegas et al., 2022; Singer et al., 2022; Khachatryan et al., 2023) have demonstrated strong capabilities of synthesizing high-quality videos following the given prompts, it is often desirable to perform adaptation for specialized tasks, such as customizing video generation to specific subjects or styles.

DreamBooth (Ruiz et al., 2023) finetunes text-to-image diffusion models to connect a unique identifier to a subject of interest, using a few images of that specific subject. The subject will be implanted into the output space of the diffusion model after finetuning, enabling novel view synthesis with the subject via prompting with its corresponding identifier. In DreamVideo (Wei et al., 2024), this

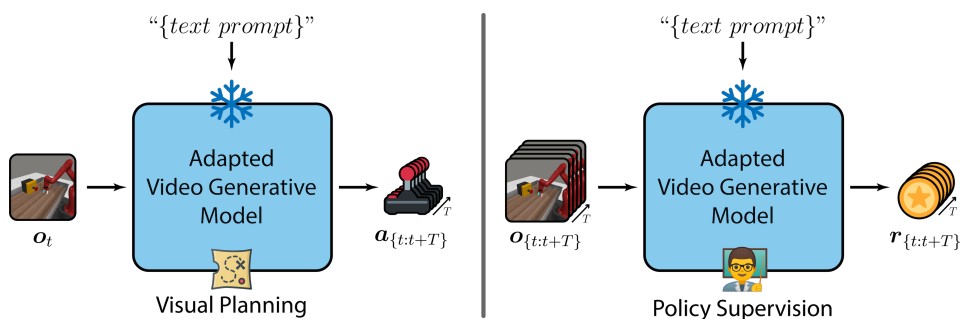

Figure 2: **Downstream Task Evaluation.** We identify downstream robotic task performance as a way to achieve standardized, quantitative comparisons across adaptation techniques. We evaluate how adapted video models can enable text-conditioned generalization via two approaches: **visual planning** and **policy supervision**. For visual planning, the adapted video model synthesizes a text-conditioned video plan into the future, which is then converted into actions to follow through a separately trained inverse dynamics model. In policy supervision, the adapted video model is used in a discriminative manner to evaluate frames achieved by the policy; these are converted into text-conditioned rewards, which the policy is optimized to maximize.

idea is extended to facilitate novel video generation with respect to a particular subject of interest. It learns subject customization for a pretrained video diffusion model through a few provided static images, which is achieved by combining textual inversion with finetuning an identity adapter.

Prior work on large-to-small adaptation of video models, through composing predicted scores, has demonstrated successful transfer of artistic styles while maintaining powerful text-conditioning behavior (Yang et al., 2023a). In this work, we evaluate this approach to explore the degree to which in-domain environment dynamics and notions of expert behaviors similarly generalize through adaptation with large-scale pretrained video models, conditioned flexibly on natural language. Furthermore, we propose and evaluate a novel probabilistic adaptation technique, which performs score composition in an inverted manner from that presented in (Yang et al., 2023a).

Prior adaptation works mostly seek improvements over visual quality and its related metrics, such as FID (Heusel et al., 2017) and FVD (Thomas et al., 2018). Here, we focus on a new application domain to evaluate adaptation: robotic task performance. We study how the text-conditioning capabilities of large-scale pretrained video generative models can be combined with environment-specific information to deliver further improvements on text-conditioned task generalization.

**Video Models for Decision Making.** A large body of recent work has explored how video models may be used for decision making (Yang et al., 2024; McCarthy et al., 2024). One line of work explores how video generative models can provide rewards, particularly through a pixel interface (Sermanet et al., 2016; Ma et al., 2022). In VIPER (Escontrela et al., 2024), a video model is trained on expert demonstrations that solve the task of interest; it is then utilized to provide dense rewards to supervise downstream policies by evaluating the likelihood of achieved frames during interaction. Similarly, expert demonstrations are also used in Diffusion-Reward (Huang et al., 2023), but a diffusion model is trained instead. Rewards are once again provided through achieved frames, but through a novel cross-entropy computation. In Video-TADPoLe (Luo et al., 2024), a large-scale pretrained video diffusion model is used to provide text-conditioned rewards through achieved environment-rendered frames. In this work, we also seek to use video generative models as supervisors for policy learning, but we treat it as a method to evaluate the efficacy of different techniques for adapting large-scale pretrained video models to in-domain data.

A separate line of work utilizes video models as pixel-based planners (Ko et al., 2024; Du et al., 2024a;b; Ajay et al., 2023; Wen et al., 2023; Liang et al., 2024; Yang et al., 2023b; Zhou et al., 2024b; Wang et al., 2024a; Zhou et al., 2024a). In such works, the video model can be directly used to generate a visual plan to solve a task, which can be converted into actions using an inverse dynamics model (Du et al., 2024a) or through dense 3D correspondences (Ko et al., 2024). Alternatively, the video model can also be used as a visual dynamics model as part of a more complex planning routine (Ajay et al., 2023; Du et al., 2024b), to form more complex, long horizon video plans. We utilize video models as visual planners for robotic tasks to understand the quality of different adaptation techniques in integrating in-domain data into large-scale pretrained text-to-video models.

## 3 Method

Video models pretrained on internet-scale data exhibit strong zero-shot generalization capabilities across diverse visual scenarios, which make them attractive to leverage for downstream robotic tasks. However, the general nature of their pretraining may not inherently enable them to understand domain-specific nuances of the environment within which we would like to learn robotic behavior. We investigate how this can be addressed by integrating in-domain information into large-scale text-to-video models; in Section 3.1 we describe three adaptation approaches of interest, each with separate requirements on in-domain training data and optimization cost. Then, in Section 3.2, we describe two techniques through which we can evaluate the novel task generalization capabilities of these adapted video models in a standardized manner.

### 3.1 Adaptation Techniques

We aim to adapt pretrained large video models to in-domain data, while maintaining and generalizing their internet-scale knowledge to solving downstream robotic tasks. We utilize an AnimateDiff (Guo et al., 2023) checkpoint as our large video model, which is designed to effectively animate pretrained text-to-image diffusion models such as StableDiffusion (Rombach et al., 2022). At its core, AnimateDiff features a motion module pretrained on a large-scale video dataset, providing powerful motion priors to guide video generation. We investigate three different techniques for in-domain adaptation: ***direct finetuning***, ***subject customization***, and ***probabilistic adaptation***.

#### 3.1.1 Direct Finetuning

Directly finetuning a generally-pretrained text-to-video model is one of the most straightforward ways to mitigate potential domain gaps. Given a video $\tau_0$ sampled from in-domain data distribution $p(\tau_0)$, a randomly sampled Gaussian noise $\epsilon \sim \mathcal{N}(\mathbf{0}, \mathbf{I})$ and a schedule of noise levels $\beta_t$ that are indexed by timestep $t \in [0, T]$, a text-conditioned video diffusion model can be trained or finetuned as a denoising function $\epsilon_\theta(\cdot)$ by optimizing the denoising objective (Ho et al., 2020) below:

$$\mathcal{L}_{\text{denoise}}(\theta) = \mathbb{E}_{\tau_0, \epsilon, t}[||\epsilon - \epsilon_\theta(\tau_t, t \mid \text{text})||^2] \tag{1}$$

in which $\tau_t$ is a noise corrupted video obtained by perturbing $\tau_0$ with sampled Gaussian noise $\epsilon$ and noise level $t$, and training is done with text-conditioning dropout to implement classifier-free guidance (Ho & Salimans, 2022). AnimateDiff is implemented as a trained motion module built on top of pretrained StableDiffusion components; in our study we keep these reused parts unchanged and only adjust the motion module. Direct finetuning involves additional training with labelled pairs of video demonstrations collected from the domain of interest, allowing the model to update the parameters of its motion module to arbitrary extents and shift its output space towards the target domain. However, direct finetuning of large video models may cause issues such as model collapse or catastrophic forgetting to emerge, especially when demonstration samples are limited.

#### 3.1.2 Subject Customization

Performing direct finetuning on pretrained large video models can often be a computationally expensive endeavour. Furthermore, it requires labelled in-domain video demonstrations, the ready availability of which cannot always be assumed in adaptation scenarios of interest for downstream robotic tasks. We therefore investigate cheaper alternatives for adaptation with respect to data and optimization cost. Customized generation (Ruiz et al., 2023; Gal et al., 2023; Wei et al., 2024) has been widely used for synthesizing subjects and scenes that accommodate user preferences, utilizing only a few static images of a subject. In this work, we also explore how this technique can be used to inject subject-centric information into pretrained video models, potentially enabling them to better supervise robotic task behavior. We use DreamBooth (Ruiz et al., 2023) to customize the generation process due to its simplicity and data efficiency. This method binds a unique text identifier to a specific subject, examples of which are provided using still images paired with text captions *without* motion information, enabling novel view synthesis of the subject contextualized in different scenes.

Following DreamBooth (Ruiz et al., 2023), we design special prompts by using a rare token (e.g. "[D]") as the unique identifier for each in-domain subject (e.g."a photo of [D] robot arm"). We finetune StableDiffusion with DreamBooth on static images of the subject paired with this special text

prompt. We then instantiate AnimateDiff with the DreamBooth-finetuned U-Net and text encoder for subject-informed video generation. Unlike direct finetuning which requires labelled video data, this approach performs few-shot customization just through the use of static images; the adaptation is possible without requiring expert video demonstrations, when only still observations of the environment and its subject are available. Since this adaptation technique will not expose any subject motions to the video model, it also allows us to study whether the pretrained video model can directly transfer its *motion prior*, which is obtained from domain-agnostic pre-training, onto arbitrary in-domain subjects of interest and facilitate generalization over downstream robotic tasks.

### 3.1.3 PROBABILISTIC ADAPTATION

Under some circumstances, the large-scale pretrained model is available for inference, but adjusting it in any way may not be feasible or desirable. In such scenarios, we consider Probabilistic Adaptation (Yang et al., 2023a), where a small sample of demonstrations is utilized to train an in-domain video model, which can be flexibly parameterized to accommodate available modeling resource constraints. Adaptation is then performed by combining the predicted scores from the pretrained, frozen large-scale video model $\epsilon_{\text{pretrained}}(\tau_t, t \mid \text{text})$ with those of the lightweight domain-specific video model $\epsilon_\theta(\tau_t, t \mid \text{text})$ during inference. We use low-temperature sampling (Yang et al., 2023a) to compute the adapted score following the denoising function below:

$$\tilde{\epsilon} = \epsilon_\theta(\tau_t, t) + \alpha\Big(\epsilon_\theta(\tau_t, t \mid \text{text}) + \gamma\epsilon_{\text{pretrained}}(\tau_t, t \mid \text{text}) - \epsilon_\theta(\tau_t, t)\Big) \tag{2}$$

where $\gamma$ is the prior strength, and $\alpha$ is the guidance scale of text-conditioning. This method only requires the training of a small component with limited in-domain data, and allows the pretrained large video model to serve as a probabilistic prior which guides the generation process of the domain-specific model through score composition.

Moreover, we extend Equation 2 to its inverse version, in which the adaptation direction between $\epsilon_\theta(\tau_t, t \mid \text{text})$ and $\epsilon_{\text{pretrained}}(\tau_t, t \mid \text{text})$ is inverted:

$$\tilde{\epsilon}_{\text{inv}} = \epsilon_{\text{pretrained}}(\tau_t, t) + \alpha\Big(\epsilon_{\text{pretrained}}(\tau_t, t \mid \text{text}) + \gamma\epsilon_\theta(\tau_t, t \mid \text{text}) - \epsilon_{\text{pretrained}}(\tau_t, t)\Big). \tag{3}$$

In inverse probabilistic adaptation, the pretrained video model controls the generation process, while consulting the small model for domain-specific information. Both probabilistic adaptation formulations allow more flexible and low-cost adaptation in the video space compared to direct finetuning; empirically, we find that one direction may work better than the other in certain circumstances.

### 3.2 EVALUATING TASK GENERALIZATION CAPABILITIES OF VIDEO MODELS

To measure the quality of adaptation, samples from the adapted video models can be judged with respect to in-domain examples in terms of Fréchet Video Distance (FVD) scores (Yang et al., 2023a; Thomas et al., 2018). However, beyond simply assessing surface-level visual style, we propose further evaluating adaptation quality via their ability to facilitate downstream robotic performance. For tasks with predefined evaluation schemes, this provides a quantifiable metric in terms of achieved performance and success, and can deliver additional insights into the capabilities of video models beyond appealing visual content generation. In this work we consider two approaches, depicted in Figure 2, for applying video models to decision making – ***visual planning*** and ***policy supervision***. Under both scenarios, we can measure to what degree downstream robotic performance and text-conditioned generalization may be enabled via different adaptation techniques.

#### 3.2.1 VIDEO MODELS AS VISUAL PLANNERS

One intuitive approach to use video generative models for decision-making is to use synthesized videos as a visual plan of the future, which then can be converted into executable actions. Concretely, a planning model learns $p(s' \mid s)$, for state $s$ and subsequent state $s'$; in the case of visual planning, $s$ takes the form of RGB frames and $p(s' \mid s)$ is a video generative model. Synthesized video plans are then translated into actions to execute in the environment through a separately learned inverse dynamics model $p(a \mid s, s')$. Applying text-guided video generation for task planning has been successfully applied in prior work (Du et al., 2024a;b; Ajay et al., 2023). However, as the performance of video planning can be highly dependent on both the visual quality of the imagined

plan and the robustness of the inverse dynamics model, prior work has only utilized video models trained on *in-domain demonstrations*.

In this work we identify and exploit the fact that the planning model $p(s' \mid s)$ does not require any action information in its training, and is therefore a promising component through which information from large-scale data and pretraining can be introduced. Through the lens of visual planning, we explicitly examine integrating large-scale video information into the planner $p(s' \mid s)$ by directly leveraging a video generative model pretrained on internet-scale data.

A set of action-labeled trajectories is still required to train an inverse dynamics model; in this work we also utilize a small set of such demonstration data to temper the large-scale pretrained video model to generate coherent in-domain visual plans while preserving priors summarized from large-scale pretraining. We therefore investigate whether generally-pretrained models can capture environment-specific dynamics through cheap adaptation to in-domain data while leveraging preserved text-conditioned generalization capabilities to facilitate performant in-domain visual planning even for *novel* tasks of interest unseen during adaptation.

### 3.2.2 VIDEO MODELS AS POLICY SUPERVISORS

In addition to their use as planners, video generative models can be used as policy supervisors. In this approach, the video model is utilized to evaluate frames achieved by the agent during interaction in a discriminative manner; these signals can then be converted into rewards with which to optimize the policy. While many prior works require video models that are trained solely on environment-specific demonstrations of expert quality (Huang et al., 2023; Escontrela et al., 2023), with rewards computed against a summarized notion of "expertness", here we seek to extract accurate text-conditioned rewards from text-to-video models. Video-TADPole (Luo et al., 2024) measures text-alignment of robotic trajectories by noise-corrupting achieved pixel observations, and evaluating how likely a large-scale pretrained text-to-video model would reconstruct the video interactions conditioned on the provided natural language prompt (additional details provided in Appendix C). We train a text-conditioned policy by maximizing cumulative Video-TADPoLe rewards through reinforcement learning. Successful optimization of such a policy enables us to evaluate the ability of adapted large-scale text-to-video models in facilitating novel task generalization, specified by natural language.

After adapting to limited data samples, synthesizing coherent high-quality in-domain videos can still be challenging for video models. This may pose issues in visual planning, where the generated plan must appear sufficiently in-domain for an inverse dynamics model to be able to accurately translate into meaningful actions. On the contrary, video models do not necessarily need the ability to create high-quality in-domain videos from scratch to behave as effective policy supervisors; they simply need to be able to critique the quality of achieved in-domain frames. Thus, expressing adapted knowledge through rewards may allow the detachment of downstream policy performance from demands on video generation quality. Conversely, a potential drawback of this approach in comparison to visual planning is the high variance commonly observed in the policy learning process.

## 4 EXPERIMENTS

### 4.1 EXPERIMENTAL SETUP AND EVALUATION

**Benchmarks:** We evaluate to what degree adapted video models can facilitate downstream robotic behavior generalization across a variety of environments and tasks, spanning robotic manipulation to continuous control. We focus the bulk of our explorations on MetaWorld-v2 (Yu et al., 2020), which offers a suite of robotic manipulation tasks with different levels of complexity. This benchmark allows us to thoroughly assess the generalization capabilities of adaptation methods across a wide selection of tasks. To study the effectiveness of adaptation techniques in a *low data* regime, we curate a small dataset of in-domain examples from 7 MetaWorld tasks (denoted with an asterisk in Table A1) to adapt pretrained video models. For each task, we utilize 25 expert videos for direct finetuning and probabilistic adaptation, while sampling a small set of non-consecutive observations for subject customization. During inference, we evaluate the adapted video models on 9 tasks, 7 of which are novel tasks that are not exposed during adaptation (denoted with no asterisk in Table A1).

| Episode Return | Vanilla AnimateDiff | In-Domain-Only | Direct Finetuning | Subject Customization | Prob. Adaptation | Inverse Prob. Adaptation |
|---|---|---|---|---|---|---|
| Humanoid Walking | $145.8 \pm 48.2$ | $2.4 \pm 0.3$ | $111.5 \pm 106.4$ | $174.7 \pm 42.7$ | $1.8 \pm 0.2$ | $92.6 \pm 51.1$ |
| Dog Walking | $60.2 \pm 8.8$ | $76.2 \pm 29.5$ | $44.6 \pm 44.3$ | $117.9 \pm 49.7$ | $11.3 \pm 0.9$ | $88.7 \pm 9.0$ |
| Overall | 103 | 39.3 | 78.1 | **146.3** | 6.5 | 90.7 |

Table 1: **Policy Supervision on Continuous Control.** We report ground-truth episode return achieved by policies optimized using the listed adapted video models, aggregated over 5 seeds. We observe that direct finetuning produces marginal improvement over a vanilla AnimateDiff model, and surprisingly, despite adaptation on just static images, subject customization is able to substantially improve continuous locomotion performance over the base pretrained video model.

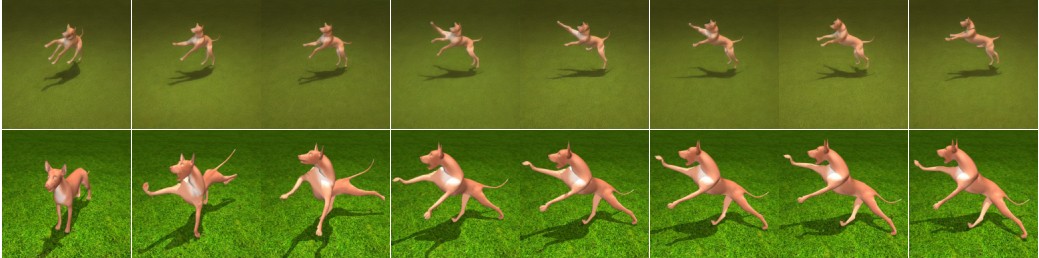

Figure 3: **Novel Text-Conditioned Generalization.** In the top row, we visualize a free-form video generation from a directly finetuned AnimateDiff model for the novel text prompt "a dog jumping". This was a behavior unseen during adaptation. When using this adapted video model for policy supervision, we showcase that it can successfully supervise a downstream Dog agent to behave according to novel text specifications in a zero-shot manner (policy rollout shown in bottom row).

Additionally, we extend our evaluation to Humanoid and Dog environments from the DeepMind Control Suite (Tassa et al., 2018). We select "Dog walking" and "Humanoid walking", which offer quantitative evaluation through ground-truth rewards, as tasks where we collect 20 demonstrations for adaptation. Following Video-TADPoLe (Luo et al., 2024), we evaluate the adapted models on walking as well as other behavior achievement tasks specified by novel text prompts. We provide a detailed list of text prompts used for both adaptation and task evaluation in Table A1.

**Implementation details of adaptation:** In our experiments, we use AnimateDiff (Guo et al., 2023) ($\sim$1.5B parameters) as our pretrained text-to-video model, which combines StableDiffusion with a motion module pretrained on WebVid-10M (Bain et al., 2021) for high-quality video generation. To perform direct finetuning on AnimateDiff, we follow the training pipeline provided by the authors and only update the parameters of its motion module with a small in-domain dataset. For subject customization, we utilize 20 static images to finetune StableDiffusion with DreamBooth for each environment. In addition, we adopt DreamBooth LoRA (Hu et al., 2022) for lower memory usage and better training efficiency. In probabilistic adaptation, we implement our small in-domain video model based on AVDC (Ko et al., 2024), a text-to-video model that diffuses over pixel space; implemented using $\sim$109M parameters, this is comparable in size to that of the small models used in prior work (Yang et al., 2023a). To enable direct score composition between the in-domain model and AnimateDiff, we modify the AVDC model to diffuse over the same latent space used by StableDiffusion. We include detailed hyperparameters for in-domain model training in Appendix D.

**Evaluation metrics:** For robotic manipulation tasks in MetaWorld, we report the "success rate", computed as the proportion of evaluation rollouts in which the agent successfully completes the given task. For Dog and Humanoid evaluation, we follow the setup in Video-TADPoLe (Luo et al., 2024) to report quantitative performance for walking tasks, which have ground-truth reward functions, while providing qualitative results for novel behavior achievement where ground-truth reward functions are unavailable. As in prior work (Yang et al., 2023a), we also use FVD (Thomas et al., 2018) to measure the visual quality of videos generated by adapted video models.

### 4.2 POLICY SUPERVISION

We implement video models as policy supervisors following the setup in Video-TADPoLe (Luo et al., 2024). We reuse the same TDMPC (Hansen et al., 2022) backbone for policy optimization, along with the same hyperparameter settings and training steps, which are tabulated in Section D.

| Success Rate (%) w/ | Door Close* | Door Open | Window Close | Window Open | Drawer Close |
|---|---|---|---|---|---|
| In-Domain-Only | $100.0 \pm 0.0$ | $31.1 \pm 44.0$ | $0.0 \pm 0.0$ | $33.3 \pm 47.1$ | $74.4 \pm 36.2$ |
| Vanilla AnimateDiff | $100.0 \pm 0.0$ | $0.0 \pm 0.0$ | $33.3 \pm 47.1$ | $31.1 \pm 44.0$ | $98.9 \pm 1.5$ |
| Direct Finetuning | $100.0 \pm 0.0$ | $0.0 \pm 0.0$ | $0.0 \pm 0.0$ | $47.8 \pm 41.4$ | $95.6 \pm 7.7$ |
| Subject Customization | $100.0 \pm 0.0$ | $0.0 \pm 0.0$ | $100.0 \pm 0.0$ | $60.0 \pm 42.5$ | $100.0 \pm 0.0$ |
| Prob. Adaptation | $95.6 \pm 7.7$ | $30.0 \pm 52.0$ | $33.3 \pm 57.7$ | $100.0 \pm 0.0$ | $100.0 \pm 0.0$ |
| Inverse Prob. Adaptation | $97.8 \pm 3.8$ | $65.6 \pm 56.8$ | $98.9 \pm 1.9$ | $98.9 \pm 1.9$ | $100.0 \pm 0.0$ |
| Success Rate (%) w/ | Drawer Open | Coffee Push* | Soccer | Button Press | **Overall** |
| In-Domain-Only | $0.0 \pm 0.0$ | $0.0 \pm 0.0$ | $0.0 \pm 0.0$ | $33.3 \pm 47.1$ | 30.2 |
| Vanilla AnimateDiff | $33.3 \pm 47.1$ | $28.9 \pm 23.2$ | $0.0 \pm 0.0$ | $33.3 \pm 47.1$ | 39.8 |
| Direct Finetuning | $0.0 \pm 0.0$ | $30.0 \pm 26.0$ | $5.6 \pm 9.6$ | $0.0 \pm 0.0$ | 31.0 |
| Subject Customization | $0.0 \pm 0.0$ | $15.6 \pm 22.0$ | $20.0 \pm 17.8$ | $0.0 \pm 0.0$ | 44.0 |
| Prob. Adaptation | $0.0 \pm 0.0$ | $1.1 \pm 1.9$ | $2.2 \pm 3.8$ | $0.0 \pm 0.0$ | 40.2 |
| Inverse Prob. Adaptation | $66.7 \pm 57.7$ | $13.3 \pm 23.0$ | $6.7 \pm 6.7$ | $66.7 \pm 57.7$ | **68.3** |

Table 2: **Policy Supervision on MetaWorld.** We report the mean success rate across 9 manipulation tasks in MetaWorld, over 3 seeds. "*" denotes seen tasks during adaptation. We observe that inverse probabilistic adaptation achieves the highest overall performance, both in averaged success rate over the entire task suite, as well as successful generalization to the highest number of novel tasks. Subject customization also achieves surprisingly high aggregate success given its cheap data cost.

**Continuous Control:** In Table 1, we report walking performance as evaluated by the ground-truth reward function across all adaptation techniques, in comparison to Video-TADPoLe using default AnimateDiff. In terms of Video-TADPoLe parameters, we utilize context window size of 8, stride length of 4 for both Humanoid and Dog experiments. We also list noise levels used for different adaptation techniques in Table A5. These settings were discovered in an offline manner using *policy discrimination*, which is described in Appendix E. Utilized text prompts can be found in Table A1.

We first observe that Direct Finetuning shows a slight decrease compared to vanilla AnimateDiff, demonstrating the challenges of capturing complex in-domain dynamics through small-scale finetuning. We also observe that the small in-domain model performs poorly on walking tasks. Consequently, both probabilistic adaptation and its inverse further show degradation on average performance when integrating vanilla AnimateDiff with the in-domain model. However, Inverse Probabilistic Adaptation not only surpasses the in-domain model on Humanoid Walking by a large margin, but also achieves stronger Dog Walking performance than both pretrained and in-domain models. We conjecture that while our small in-domain model has difficulty modeling Humanoid and Dog motions, increasing in-domain model capacity and applying proper adaptation with pretrained models yields improvements through Video-TADPoLe. We further discover that Subject Customization performs better on default walking behavior across both environments, with significant improvement in the case of Dog. This is a striking result, as Subject Customization only utilizes static images of the scene for adaptation and reuses the default motion module pretrained from AnimateDiff.

However, for novel text-conditioned generalization to new poses, we find that direct finetuning performs the best. For a novel text prompt and motion, such as "a dog jumping", a directly finetuned video model is able to supervise the learning of an associated policy. We provide a visual of the achieved policy rollout in Figure 3. These results indicate that for continuous locomotion settings, direct finetuning may be the best balance in terms of preserving performance but also enabling interesting text-conditioned generalization; subject customization, on the other hand, is a low-cost yet performant approach to consider. Finally, Inverse Probabilistic Adaptation enables competitive continuous control performance without finetuning the large pretrained model.

**Robotic Manipulation:** In Table 2, we report the average success rate on MetaWorld tasks across adaptation techniques, using a standardized context window of 8, stride length of 4, and noise level of 700. We discover that inverse probabilistic adaptation has the best performance. It is able to achieve non-zero success rate over all 9 tasks, with the highest average success rate of 68.3%. By default, utilizing vanilla AnimateDiff through Video-TADPoLe is able to achieve decent performance, highlighting the default text-conditioned generalization capabilities of large pretrained models. We also believe that integrating it with an in-domain model that supervises the motion of the particular dynamics of the environment, enables better generalization on more challenging tasks (e.g. Door Open). However, the default probabilistic adaptation formulation may heavily rely on the in-domain text-conditioned score, which may be inaccurate when handling novel task prompts

| Success Rate (%) w/ | Door Close* | Door Open | Window Close | Window Open | Drawer Close |
|---|---|---|---|---|---|
| In-Domain-Only | 93.3 ± 14.9 | 0.0 ± 0.0 | 53.3 ± 29.8 | 6.7 ± 14.9 | 20.0 ± 29.8 |
| Vanilla AnimateDiff | 100.0 ± 0.0 | 0.0 ± 0.0 | 13.3 ± 18.3 | 40.0 ± 27.9 | 46.7 ± 29.8 |
| Direct Finetuning | 100.0 ± 0.0 | 0.0 ± 0.0 | 0.0 ± 0.0 | 6.7 ± 14.9 | 80.0 ± 29.8 |
| Subject Customization | 100.0 ± 0.0 | 0.0 ± 0.0 | 0.0 ± 0.0 | 20.0 ± 29.8 | 25.0 ± 31.9 |
| Prob. Adaptation | 100.0 ± 0.0 | 0.0 ± 0.0 | 73.3 ± 27.9 | 13.3 ± 18.3 | 40.0 ± 43.5 |
| Inverse Prob. Adaptation | 100.0 ± 0.0 | 0.0 ± 0.0 | 53.3 ± 18.3 | 0.0 ± 0.0 | 53.3 ± 38.0 |

| Success Rate (%) w/ | Drawer Open | Coffee Push* | Soccer | Button Press | **Overall** |
|---|---|---|---|---|---|
| In-Domain-Only | 0.0 ± 0.0 | 0.0 ± 0.0 | 0.0 ± 0.0 | 40.0 ± 14.9 | 23.7 |
| Vanilla AnimateDiff | 0.0 ± 0.0 | 0.0 ± 0.0 | 0.0 ± 0.0 | 0.0 ± 0.0 | 22.2 |
| Direct Finetuning | 0.0 ± 0.0 | 13.3 ± 18.2 | 6.7 ± 14.9 | 0.0 ± 0.0 | 23.0 |
| Subject Customization | 0.0 ± 0.0 | 13.3 ± 29.8 | 0.0 ± 0.0 | 0.0 ± 0.0 | 17.6 |
| Prob. Adaptation | 6.7 ± 14.9 | 6.7 ± 14.9 | 0.0 ± 0.0 | 33.3 ± 23.6 | **30.4** |
| Inverse Prob. Adaptation | 0.0 ± 0.0 | 0.0 ± 0.0 | 0.0 ± 0.0 | 26.7 ± 27.9 | 25.9 |

Table 3: **Visual Planning on MetaWorld.** We report the mean success rate via visual planning across 9 tasks, aggregated over 5 seeds each. We discover that both probabilistic adaptation and its inverse are able to act as performant visual planners, and substantially improve over vanilla AnimateDiff; notably, probabilistic adaptation achieves success on more unseen tasks than alternatives.

due to its small training scale. We therefore hypothesize that this explains why inverse probabilistic adaptation is more performant; it may be more robust to novel text-conditioning, as more weight is put on leveraging textual priors from the pretrained model.

## 4.3 VISUAL PLANNING

We implement video models as visual planners following the framework in UniPi (Du et al., 2024a). To generate a plan, we synthesize a sequence of 8 future frames conditioned on both the current visual observation from the environment and the text prompt specifying the task. This is then translated into an executable action sequence via an inverse dynamics model. To mitigate the potential error accumulation problem, we evaluate our visual planner in a closed-loop manner, in which we only execute the first inferred action for every environment step. We provide detailed hyperparameters for video planning, and the implementation of the inverse dynamics model, in Appendix D.

**Robotic Manipulation:** We evaluate visual planners with different adaptation techniques across 9 selected tasks in MetaWorld, of which 7 are unseen during adaptation, and report the success rates in Table 3. Among all evaluated adaptation techniques, we observe that probabilistic adaptation and its inverse version achieve the highest overall performance, with probabilistic adaptation achieving the highest number of non-zero performance on novel tasks. We find that using only static images for adaptation through subject customization fails to produce useful in-domain visual plans. Meanwhile, direct finetuning only brings marginal improvement over Vanilla AnimateDiff and performs on par with the small in-domain model, struggling to generalize well on unseen tasks due to the small scale of demonstration data being considered. For such settings with cheap amounts of in-domain data, we discover that probabilistic adaptation and its inverse appear comparatively more promising.

**Additional Metrics:** Visual planning relies on synthesizing coherent high-quality video plans that can be accurately interpreted by the inverse dynamics model. Therefore, we may be interested in measuring the quality of visual plans created by adapted video models in a free-form manner, in other words, generated from pure noise. In Table 4, we report FVD scores of free-form generation for both seen and unseen tasks in MetaWorld benchmark. In each setup, we generate 1,000 synthetic videos over 7 robotic manipulation tasks. We observe that probabilistic adaptation and its inverse have strong FVD scores compared to other adaptation techniques. At the same time, high FVD scores achieved by direct finetuning fails to translate to high task success rate through visual planning; this suggests that while its plans are of high visual quality, they may not necessarily model the correct motions for solving tasks in a text-conditioned manner. This further suggests that beyond just gauging visual metrics alone, using downstream robotic tasks can provide meaningful and additional insights to evaluate adaptation performance of video models.

**Studying Data Quality:** In probabilistic adaptation, our in-domain model is trained solely on limited expert demonstrations, which can still be prohibitively expensive to collect in some scenarios. By combining with pretrained video models through adaptation, we provide further investigation on whether the knowledge (e.g. motion priors) obtained from large-scale pretraining can bridge the gap between the suboptimality of in-domain data and task evaluation performance.

| FVD Scores (MetaWorld) | Vanilla AnimateDiff | In-Domain-Only | Direct Finetuning | Subject Customization | Prob. Adaptation | Inverse Prob. Adaptation |
|---|---|---|---|---|---|---|
| Seen | 4625.3 | 2987.2 | 946.0 | 2212.8 | **848.3** | 928.6 |
| Unseen | 4469.7 | 3080.7 | **915.0** | 2316.0 | 1237.6 | 1250.3 |

Table 4: **FVD Scores with Free-form Generation.** We report FVD scores for videos of MetaWorld tasks, produced by Free-form Generation via the video generative models of interest. This is computed for both seen and unseen task sets, each with 7 tasks, aggregated over 1000 synthetic videos.

| Success Rate (%) w/ | Door Close* | Door Open | Window Close | Window Open | Drawer Close |
|---|---|---|---|---|---|
| In-Domain-Only | $93.3 \pm 14.9$ | $0.0 \pm 0.0$ | $40.0 \pm 27.9$ | $0.0 \pm 0.0$ | $33.3 \pm 23.6$ |
| Prob. Adaptation | $100.0 \pm 0.0$ | $0.0 \pm 0.0$ | $60.0 \pm 36.5$ | $13.3 \pm 18.3$ | $46.7 \pm 29.8$ |
| Inverse Prob. Adaptation | $93.3 \pm 14.9$ | $0.0 \pm 0.0$ | $53.3 \pm 29.8$ | $0.0 \pm 0.0$ | $93.3 \pm 14.9$ |

| Success Rate (%) w/ | Drawer Open | Coffee Push* | Soccer | Button Press | **Overall** |
|---|---|---|---|---|---|
| In-Domain-Only | $0.0 \pm 0.0$ | $0.0 \pm 0.0$ | $0.0 \pm 0.0$ | $13.3 \pm 18.3$ | 20.0 |
| Prob. Adaptation | $6.7 \pm 14.9$ | $0.0 \pm 0.0$ | $0.0 \pm 0.0$ | $0.0 \pm 0.0$ | 25.2 |
| Inverse Prob. Adaptation | $6.7 \pm 14.9$ | $0.0 \pm 0.0$ | $0.0 \pm 0.0$ | $0.0 \pm 0.0$ | **27.4** |

Table 5: **Probabilistic Adaptation with Suboptimal Data for Video Planning.** We report the mean success rate via visual planning across 9 tasks, aggregated over 5 seeds each. Compared to results in Table 3, we discover that the overall performance of the in-domain model and probabilistic adaptation is severely impacted by the suboptimality of the training data. In contrast, the performance of inverse probabilistic adaptation remains robust under the suboptimal setup.

In Table 5, we perform planning with probabilistic adaptation and its inverse, where the available adaptation data is produced by a suboptimal agent. The suboptimal agent takes an expert action only 30% of the time, and a random action 70% of the time. In consistency with the previous setup, we collect 25 (now suboptimal) demonstrations from the same 7 tasks denoted with asterisks in Table A1. We observe that the performance of probabilistic adaptation decreases when integrating with a suboptimal in-domain model. Surprisingly, the overall average task success rate remains robust for inverse probabilistic adaptation. Overall, both adaptation techniques outperform the in-domain only baseline. This is a promising sign that in adapting large-scale text-to-video models for robotic downstream tasks, expert demonstrations may not be explicitly needed. This potentially opens up opportunities for applying large-scale video models to novel robotic tasks, where only random or suboptimal demonstrations are tractably available.

# 5 CONCLUSION AND FUTURE WORK

In this work, we investigate how internet video knowledge can be adapted with a small amount of in-domain video demonstrations to achieve novel task generalization for downstream robotics, which we generally acronymize as Adapt2Act. We explore several methods through which internet-scale video models may be adapted to model the appearance and dynamics of novel environments, and subsequently utilized to perform new behaviors or accomplish unseen tasks conditioned on natural language. Our considered adaptation techniques vary in their data and resource requirements. We have conducted extensive evaluations on MetaWorld and DeepMind Control Suite tasks under both policy supervision and visual planning setups. In particular, our proposed *Inverse Probabilistic Adaptation* approach demonstrates strong generalization capabilities across different task settings, and remains robust when only suboptimal demonstrations are available. Our findings highlight the promise of leveraging internet-scale text-to-video priors to model domain-specific robotic behaviors through data-efficient adaptation, which enables further advancements on text-conditioned generalization for embodied intelligence.

**Limitations and Future Work.** When the same adapted model is utilized to solve the tasks under different setups, performance differences that are difficult to explain may arise. For example, while inverse probabilistic adaptation outperforms its inverse version significantly under the policy supervision setup, both achieve similar success rates under the video planning setup. Additionally, a natural next step to further study is the use of suboptimal data, which can be collected cheaply by running random policies on tasks of interest, or iteratively augmented from on-policy observations.

**Acknowledgments.** This work is supported by Samsung and NASA. Our research was conducted using computational resources at the Center for Computation and Visualization at Brown University. Chen would like to thank Mia for inspiration.

## 6 REPRODUCIBILITY

All adaptation techniques in our work are implemented using available open-sourced components. As mentioned in Section 4.1, we use publicly available checkpoints for pretrained large models, such as AnimateDiff (Guo et al., 2023) as well as StableDiffusion (Rombach et al., 2022). We utilize DreamBooth (Ruiz et al., 2023) for subject customization. Furthermore, we reuse the codebase provided by the authors of AVDC (Ko et al., 2024) for in-domain model training, with minimal adjustments to enable the latent diffusion. For policy learning, we follow Video-TADPoLe (Luo et al., 2024) framework, which itself is built off of publicly available AnimateDiff and TDMPC (Hansen et al., 2022). Furthermore, we release detailed hyperparameter settings and implementation details in Appendix D, as well as elaborate on techniques on how they were discovered or selected in Appendix E. We believe that the simplicity of our approach, along with the utilization of open-sourced checkpoints, makes this work highly reproducible. We also commit to open-sourcing our code, to support further reproducibility efforts in the community.

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

## A  TEXT PROMPTS

Below we listed the text prompts we used for adaptation and task evaluation.

| Task | In-Domain Prompts | AnimateDiff Prompts | DreamBooth Identifier |
|---|---|---|---|
| Dog Walking | a dog/pharaoh hound walking | a dog/pharaoh hound walking | a [D] dog |
| Humanoid Walking | a(n) humanoid/action figure walking | a(n) humanoid/action figure walking | a [D] action figure |
| Assembly* | assembly | a robot arm placing a ring over a peg | |
| Dial Turn* | dial turn | a robot arm turning a dial | |
| Reach* | reach | a robot arm reaching a red sphere | |
| Peg Unplug Side* | peg unplug side | a robot arm unplugging a gray peg | |
| Lever Pull* | lever pull | a robot arm pulling a lever | |
| Coffee Push* | coffee push | a robot arm pushing a white cup towards a coffee machine | |
| Door Close* | door close | a robot arm closing a door | a [D] robot arm |
| Door Open | door open | a robot arm opening a door | |
| Window Close | window close | a robot arm closing a window | |
| Window Open | window open | a robot arm opening a window | |
| Drawer Close | drawer close | a robot arm closing a drawer | |
| Drawer Open | drawer open | a robot arm open a drawer | |
| Soccer | soccer | a robot arm pushing a soccer ball into the net | |
| Button Press | button press | a robot arm pushing a button | |

Table A1: **Task-Prompt Pairs.** We include a comprehensive list of tasks and their text prompts for adaptation and evaluation. "*" denotes tasks seen during adaptation.

| Task | In-Domain Prompts | AnimateDiff Prompts |
|---|---|---|
| Spatula in Kitchen* | spatula | find the spatula |
| Toaster in Kitchen* | toaster | find the toaster |
| Painting in Living Room* | painting | find the painting |
| Blinds in Bedroom* | blinds | find the blinds |
| ToiletPaper in Bathroom* | toilet paper | find the toilet paper |
| Pillow in Living Room | pillow | find the pillow |
| DeskLamp in Living Room | desk lamp | find the desk lamp |
| Mirror in Bedroom | mirror | find the mirror |
| Laptop in Bedroom | laptop | find the laptop |

Table A2: **Task-Prompt Pairs for iTHOR.** We include a comprehensive list of iTHOR tasks and their text prompts for adaptation and evaluation. "*" denotes tasks seen during adaptation.

## B  CONTINUED DENOISING

In diffusion-based policy supervision, rewards are extracted from the procedure of corrupting frames achieved by the policy with some level of Gaussian noise and then making denoising predictions using the video model (Huang et al., 2023; Luo et al., 2024). For additional insight, we propose a visualization technique called ***continued denoising***, and report FVD scores for videos generated in such a manner. In continued denoising, rather than extracting a scalar from components of the denoising prediction as in Video-TADPoLe, we treat the noised video as an initialization and iteratively continue sampling to produce a final clean video prediction - thus, "continuing" the denoising procedure. In our experiments we perform continued denoising conditioned on a desired text prompt, a noise level of 700, a total frame length of 16, and 10 denoising steps.

As mentioned in Section 3.2.2, policy supervision does not necessarily require strong free-form generation of in-domain videos; rather it evaluates observed frames achieved by following the current policy. For qualitative purposes, continued denoising provides us a visual sense of how this evaluation of achieved frames is done (examples in Figure A1), as well as a sanity check on the integration of in-domain information through adaptation. Furthermore, it enables quantitative comparison through FVD scores, which provides an idea on the capability of adapted video models to reconstruct in-domain-like videos conditioned on text. It is intuitive to hypothesize that a lower FVD score correlates with better in-domain adaptation, as it understands how to accurately complete the provided in-domain frames from a heavy noise corruption.

In Table A3, we report the FVD scores for the same set of seen and unseen tasks that are evaluated in free-form generation experiments. We discover that the lowest FVD score for continued denoising is achieved by the in-domain model, which is unsurprising as it was explicitly trained on such examples. The next-best FVD scores are achieved by probabilistic adaptation and its inverse. This is significant because it supports the finding that with adaptation, generalization to unseen tasks is possible, and suggests that accurate domain-specific rewards can be supplied through policy supervision. Indeed, this aligns with our result in Table 2, where Inverse Probabilistic Adaptation achieves the best overall task performance through policy supervision.

| FVD Scores (MetaWorld) | Vanilla AnimateDiff | In-Domain-Only | Direct Finetuning | Subject Customization | Prob. Adaptation | Inverse Prob. Adaptation |
|---|---|---|---|---|---|---|
| Seen | 2700.4 | **602.8** | 1004.6 | 1078.9 | 622.6 | 627.4 |
| Unseen | 2643.2 | **610.1** | 978.5 | 1711.8 | 630.6 | 681.8 |

Table A3: **FVD Scores with Continued Denoising.** We report FVD scores for videos of Meta-World tasks, produced by Continued Denoising via the video generative models of interest. This is computed for both seen and unseen task sets, each with 7 tasks, aggregating results over 1000 synthetic videos.

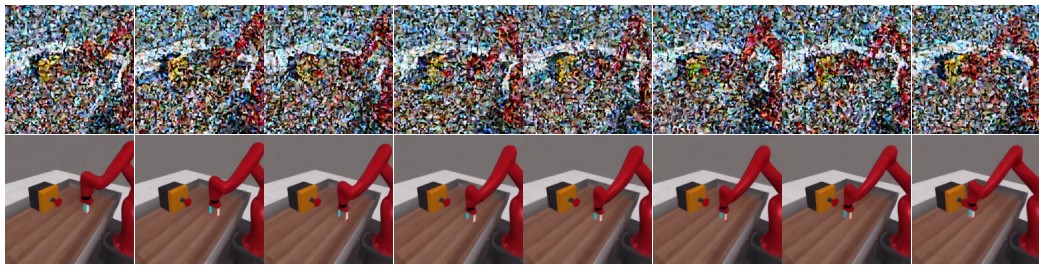

Figure A1: **Continued Denoising.** We visualize frames from a task unseen during adaptation, corrupted with a level of Gaussian noise (top row). We then show the result of continued denoising using an inverse probabilistic adaptation model to verify it can visually generalize to fill in novel in-domain information. Despite not having seen a button, it is able to reconstruct it conditioned on text. This figure is for intuition; in practice, a much higher noise level is used, shown in Figure A2.

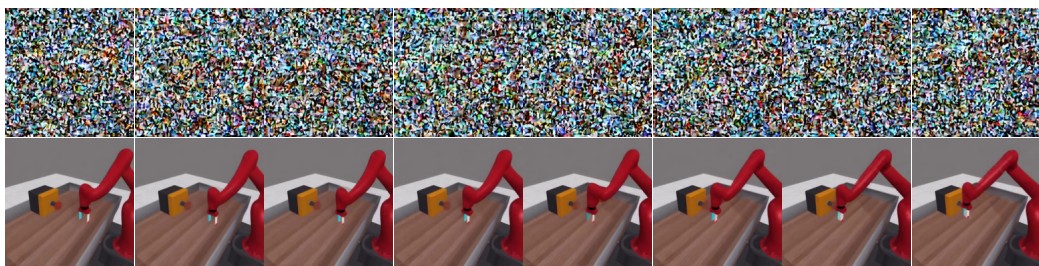

Figure A2: **Continued Denoising (in practice).** In practice, an aggressive level of Gaussian corruption is usually used on achieved frames for reward computation (700 for MetaWorld). However, because to the human eye this may look virtually indistinguishable from pure noise, we supply an illustrative example in Figure A1 using a noise level of 400. Here, we showcase visuals of the same unseen task corrupted with a practical noise level of 700. We then show the result of continued denoising to visually verify the model integrates adapted in-domain information successfully. When performing continued denoising from such a high corruption, conditioned on the text prompt "a robot arm pushing a button", it is therefore quite surprising the level of detail with which the adapted text-to-video model is able to reconstruct novel in-domain features such as the button - which it has not even seen during adaptation. The resulting continued denoising video can also be evaluated against in-domain examples via FVD for further insights.

## C   VIDEO-TADPOLE REWARD COMPUTATION

Video-TADPoLe (Luo et al., 2024) rewards are densely computed for a trajectory achieved by a policy, in terms of their rendered frames. For arbitrary start index $i$ and end index $j$ inclusive of the trajectory, for $i \leq j$, let $\mathbf{o}_{[i+1:j+1]}$ denote the associated sequence of rendered frames. Video-TADPoLe then utilizes a source noise vector $\boldsymbol{\epsilon}_0 \sim \mathcal{N}(\boldsymbol{\epsilon}; \mathbf{0}, \mathbf{I}_{j-i+1})$ of the same dimensionality as a Gaussian corruption to produce noisy observation $\tilde{\mathbf{o}}_{[i+1:j+1]}$. Then, Video-TADPoLe computes a batch of *alignment reward* terms through one inference step of the text-to-video diffusion model as:

$$r_{[i:j]}^{\text{align}} = \left\| \hat{\boldsymbol{\epsilon}}_\phi(\tilde{\boldsymbol{o}}_{[i+1:j+1]}; \mathtt{t}_{\text{noise}}, y) - \hat{\boldsymbol{\epsilon}}_\phi(\tilde{\boldsymbol{o}}_{[i+1:j+1]}; \mathtt{t}_{\text{noise}}) \right\|_2^2,$$

and a batch of *reconstruction reward* terms as:

$$r_{[i:j]}^{\text{rec}} = \left\| \hat{\boldsymbol{\epsilon}}_\phi(\tilde{\boldsymbol{o}}_{[i+1:j+1]}; \mathtt{t}_{\text{noise}}) - \boldsymbol{\epsilon}_0 \right\|_2^2 - \left\| \hat{\boldsymbol{\epsilon}}_\phi(\tilde{\boldsymbol{o}}_{[i+1:j+1]}; \mathtt{t}_{\text{noise}}, y) - \boldsymbol{\epsilon}_0 \right\|_2^2.$$

For a provided context window of size $n$, Video-TADPoLe calculates the reward at each timestep $t$ utilizing each context window that involves achieved observation $\mathbf{o}_{t+1}$:

$$r_t = \frac{1}{n} \sum_{i=1}^{n} \mathtt{symlog}\left( w_1 * r_{[t-i+1:t-i+n]}^{\text{align}}[i-1] \right) + \mathtt{symlog}\left( w_2 * r_{[t-i+1:t-i+n]}^{\text{rec}}[i-1] \right).$$

A stride term $s$ can be used to make this computation tractable across long trajectories, where the context window skips $s$ timesteps before computing a sequence of Video-TADPoLe rewards again. The context window $n$, stride $s$, and noise level $\mathtt{t}_{\text{noise}}$ are hyperparameters to be set by the user; in practice, good settings for such hyperparameters can be found in an offline manner through *policy discrimination* (Section E).

## D   IMPLEMENTATION DETAILS

We include the default hyperparameters from the TD-MPC implementation in Table A9 for completeness. We do not modify the default recommended settings for both Humanoid and Dog environments, as well as the Meta-World experiments.

| Hyperparameter | Value |
|---|---|
| Training Objective | `pred_noise` |
| Number of Training Steps | 60000 |
| Loss Type | L2 |
| Learning Rate | 1e-4 |
| Beta Schedule | Linear schedule (0.0085, 0.012) |
| Timesteps | 1000 |
| EMA Decay | 0.99 |
| EMA Update Steps | 10 |

Table A4: **Hyperparameters for In-Domain Model Training.**

**Visual Planning Hyperparameters:** To generate a video plan with adapted video models, we perform DDIM (Song et al., 2021) sampling for 25 steps. We use 7.5 as the text-conditioning guidance scale for directly finetuned AnimateDiff, and use 2.5 for other adaptation techniques. Additionally, we use 0.1 as the prior strength for probabilistic adaptaion and 0.5 for its inverse version.

| Noise Level | Humanoid Walking | Dog Walking |
|---|---|---|
| In-Domain Only | 600 | 600 |
| Direct Finetuning | 700 | 700 |
| Subject Customization | 500 | 600 |
| Prob. Adaptation | 700 | 700 |
| Inverse Prob. Adaptation | 600 | 500 |

Table A5: **VideoTADPoLe Noise Levels for DeepMind Control.**

**Inverse Dynamics:** We employ a small MLP network as our inverse dynamics model. The model takes in the embeddings of two consecutive video frames, which are extracted using VC-1 (Majumdar et al., 2023), and predicts the action that enables the transition between the provided frames. We train the inverse dynamics model on a dataset comprising a mixture of expert and suboptimal trajectories rendered from the environment, using the same set of tasks and data volumn as used for adaptation. For fairness, we reuse the same dynamics model across all adaptation techniques during evaluation. We provide the detailed hyperparameters of inverse dynamics training in Table A6.

| Hyperparameter | Value |
|---|---|
| Input Dimension | 1536 |
| Output Dimension | 4 |
| Training Epochs | 20 |
| Learning Rate | 3e-5 |
| Optimizer | AdamW |

Table A6: **Hyperparamters of Inverse Dynamics Model Training**

| Component | # Parameters (Millions) |
|---|---|
| VAE (Encoder) | 34.16 |
| VAE (Decoder) | 49.49 |
| U-Net | 865.91 |
| Text Encoder | 340.39 |

Table A7: **StableDiffusion Components.** For completeness, we list sizes of the components of the StableDiffusion v2.1 checkpoint used in Video-TADPoLe experiments. The checkpoint is used purely for inference, and is not modified or updated in any way. Note that the VAE Decoder is not utilized in our framework.

| Component | # Parameters (Millions) |
|---|---|
| VAE (Encoder) | 34.16 |
| VAE (Decoder) | 49.49 |
| U-Net | 1312.73 |
| Text Encoder | 123.06 |

Table A8: **AnimateDiff Components.** For completeness, we list sizes of the components of the AnimateDiff checkpoint used in Video-TADPoLe experiments. The checkpoint is used purely for inference, and is not modified or updated in any way. Note that the VAE Decoder is not utilized in our framework.

| Hyperparameter | Value |
|---|---|
| Discount factor ($\gamma$) | 0.99 |
| Seed steps | $5,000$ |
| Replay buffer size | Unlimited |
| Sampling technique | PER ($\alpha = 0.6, \beta = 0.4$) |
| Planning horizon ($H$) | 5 |
| Initial parameters ($\mu^0, \sigma^0$) | $(0, 2)$ |
| Population size | 512 |
| Elite fraction | 64 |
| Iterations | 12 (Humanoid) |
| | 8 (Dog) |
| Policy fraction | 5% |
| Number of particles | 1 |
| Momentum coefficient | 0.1 |
| Temperature ($\tau$) | 0.5 |
| MLP hidden size | 512 |
| MLP activation | ELU |
| Latent dimension | 100 (Humanoid, Dog) |
| Learning rate | 3e-4 (Dog) |
| | 1e-3 (Humanoid) |
| Optimizer ($\theta$) | Adam ($\beta_1 = 0.9, \beta_2 = 0.999$) |
| Temporal coefficient ($\lambda$) | 0.5 |
| Reward loss coefficient ($c_1$) | 0.5 |
| Value loss coefficient ($c_2$) | 0.1 |
| Consistency loss coefficient ($c_3$) | 2 |
| Exploration schedule ($\epsilon$) | $0.5 \rightarrow 0.05$ (25k steps) |
| Planning horizon schedule | $1 \rightarrow 5$ (25k steps) |
| Batch size | 2048 (Dog) |
| | 512 (Humanoid) |
| Momentum coefficient ($\zeta$) | 0.99 |
| Steps per gradient update | 1 |
| $\theta^-$ update frequency | 2 |

Table A9: **TD-MPC hyperparameters.** We use the official implementation TD-MPC (Hansen et al., 2022) with no adjustments to the hyperparameters, but list it below for completeness. We set the number of training steps to 2 million for continuous control experiments using TD-MPC, and 700k steps for MetaWorld experiments.

# E  POLICY DISCRIMINATION

Rather than performing an expensive sweep over Video-TADPoLe hyperparameters directly by launching policy supervision experiments across each adapted video model technique, which can be expensive, we look for an offline method to determine reasonable hyperparameter settings. For

each environment, we therefore utilize an example expert quality demonstration video as well as an example poor quality demonstration video (with arbitrary quality levels in-between, if available). Then, we can perform a search over Video-TADPoLe parameters by computing Video-TADPoLe rewards for these trajectories using an adapted video model, conditioned on the task-relevant text prompt, with respect to different context window, stride, and noise level settings. We seek parameter settings that, through the adapted video model's Video-TADPoLe reward computation, can correctly distinguish between the expert, text-aligned video demonstration from the poor, text-unaligned video demonstration; this can be done by comparing the predicted Video-TADPoLe rewards. Once identified in this offline manner, we can subsequently use the discovered settings of context window, stride, and noise level for learning text-conditioned policies. In practice, we have found that these settings can be reused for novel text-conditioning within the same environment without issue.

## F  POLICY SUPERVISION WITH ADDITIONAL PRETRAINED VIDEO MODELS

| Success Rate (%) w/ | Door Close* | Door Open | Window Close | Window Open | Drawer Close |
|---|---|---|---|---|---|
| In-Domain-Only | $100.0 \pm 0.0$ | $31.1 \pm 44.0$ | $0.0 \pm 0.0$ | $33.3 \pm 47.1$ | $74.4 \pm 36.2$ |
| Vanilla AnimateLCM | $100.0 \pm 0.0$ | $0.0 \pm 0.0$ | $98.9 \pm 1.9$ | $33.3 \pm 29.1$ | $100.0 \pm 0.0$ |
| Prob. Adaptation | $100.0 \pm 0.0$ | $0.0 \pm 0.0$ | $66.7 \pm 57.7$ | $0.0 \pm 0.0$ | $100.0 \pm 0.0$ |
| Inverse Prob. Adaptation | $100.0 \pm 0.0$ | $100.0 \pm 0.0$ | $100.0 \pm 0.0$ | $94.4 \pm 9.6$ | $100.0 \pm 0.0$ |
| **Success Rate (%) w/** | **Drawer Open** | **Coffee Push*** | **Soccer** | **Button Press** | **Overall** |
| In-Domain-Only | $0.0 \pm 0.0$ | $0.0 \pm 0.0$ | $0.0 \pm 0.0$ | $33.3 \pm 47.1$ | 30.2 |
| Vanilla AnimateLCM | $0.0 \pm 0.0$ | $5.6 \pm 9.6$ | $0.0 \pm 0.0$ | $0.0 \pm 0.0$ | 37.5 |
| Prob. Adaptation | $0.0 \pm 0.0$ | $32.2 \pm 28.0$ | $4.4 \pm 7.7$ | $0.0 \pm 0.0$ | 33.7 |
| Inverse Prob. Adaptation | $16.7 \pm 29.0$ | $41.1 \pm 15.0$ | $4.4 \pm 5.1$ | $30.0 \pm 52.0$ | **65.2** |

Table A10: **Policy Learning on MetaWorld with AnimateLCM.** We report the mean success rate across 9 manipulation tasks in MetaWorld, aggregated over 3 seeds.

We also provide policy supervision results on MetaWorld with AnimateLCM (Wang et al., 2024b) in Table A10. Similar to AnimateDiff, vanilla AnimateLCM is also able to achieve decent success rates through Video-TADPoLe. Furthermore, we discover that inverse probabilistic adaptation consistently achieves the best performance with both AnimateDiff and AnimateLCM. With AnimateLCM, inverse probabilistic adaptation obtains the highest overall success rate of **65.2%**, surpassing all other evaluated video models and adaptation techniques, with non-zero success rates across all evaluated tasks.

## G  VISUAL PLANNING WITH ADDITIONAL PRETRAINED VIDEO MODELS

| Success Rate (%) w/ | Door Close* | Door Open | Window Close | Window Open | Drawer Close |
|---|---|---|---|---|---|
| In-Domain-Only | $93.3 \pm 14.9$ | $0.0 \pm 0.0$ | $53.3 \pm 29.8$ | $6.7 \pm 14.9$ | $20.0 \pm 29.8$ |
| Vanilla AnimateLCM | $100.0 \pm 0.0$ | $0.0 \pm 0.0$ | $0.0 \pm 0.0$ | $20.0 \pm 18.3$ | $40.0 \pm 27.9$ |
| Prob. Adaptation | $100.0 \pm 0.0$ | $0.0 \pm 0.0$ | $53.3 \pm 38.0$ | $0.0 \pm 0.0$ | $53.3 \pm 29.8$ |
| Inverse Prob. Adaptation | $100.0 \pm 0.0$ | $0.0 \pm 0.0$ | $40.0 \pm 14.9$ | $0.0 \pm 0.0$ | $93.3 \pm 14.9$ |
| **Success Rate (%) w/** | **Drawer Open** | **Coffee Push*** | **Soccer** | **Button Press** | **Overall** |
| In-Domain-Only | $0.0 \pm 0.0$ | $0.0 \pm 0.0$ | $0.0 \pm 0.0$ | $40.0 \pm 14.9$ | 23.7 |
| Vanilla AnimateLCM | $0.0 \pm 0.0$ | $0.0 \pm 0.0$ | $0.0 \pm 0.0$ | $0.0 \pm 0.0$ | 17.8 |
| Prob. Adaptation | $0.0 \pm 0.0$ | $0.0 \pm 0.0$ | $0.0 \pm 0.0$ | $6.7 \pm 14.9$ | 23.7 |
| Inverse Prob. Adaptation | $0.0 \pm 0.0$ | $0.0 \pm 0.0$ | $6.7 \pm 14.9$ | $26.7 \pm 27.9$ | **29.6** |

Table A11: **Visual Planning on MetaWorld with AnimateLCM.** We report the mean success rate across 9 manipulation tasks in MetaWorld. Each table entry shows the average success rate aggregated from 5 seeds.

We provide visual planning results on MetaWorld with an additional video diffusion model, AnimateLCM (Wang et al., 2024b), in Table A11. We observe both probabilistic adaptation and its inverse version bring improvements in overall success rate compared to Vanilla AnimateLCM. Specifically, inverse probabilistic adaptation achieves the best overall performance and outperforms the

in-domain-only baseline by 24.9%, reconfirming the efficacy of adaptation in improving in-domain task performance. This further demonstrates that adaptation as an approach can be applied flexibly across different backbone text-to-video models for successful downstream robotic applications.

# H VISUAL PLANNING FOR OBJECT NAVIGATION IN iTHOR ENVIRONMENTS

| Success Rate (%) w/ | Spatula in *Kitchen**  | Toaster in *Kitchen**  | Painting in *Living Room**  | Blinds in *Bedroom**  | ToiletPaper in *Bathroom**  |
|---|---|---|---|---|---|
| In-Domain-Only | $13.3 \pm 29.8$ | $33.3 \pm 33.3$ | $0.0 \pm 0.0$ | $13.3 \pm 29.8$ | $40.0 \pm 36.5$ |
| Prob. Adaptation | $20.0 \pm 29.8$ | $60.0 \pm 27.9$ | $0.0 \pm 0.0$ | $26.7 \pm 36.5$ | $73.3 \pm 14.9$ |
| Inverse Prob. Adaptation | $13.3 \pm 18.3$ | $33.3 \pm 23.6$ | $0.0 \pm 0.0$ | $33.3 \pm 33.3$ | $40.0 \pm 14.9$ |
| Success Rate (%) w/ | Pillow in *Living Room* | DeskLamp in *Living Room* | Mirror in *Bedroom* | Laptop in *Bedroom* | **Overall** |
| In-Domain-Only | $6.7 \pm 14.9$ | $6.7 \pm 14.9$ | $0.0 \pm 0.0$ | $26.7 \pm 27.9$ | 15.6 |
| Prob. Adaptation | $13.3 \pm 18.3$ | $13.3 \pm 18.3$ | $0.0 \pm 0.0$ | $53.3 \pm 29.8$ | **28.9** |
| Inverse Prob. Adaptation | $6.7 \pm 14.9$ | $13.3 \pm 18.3$ | $6.7 \pm 14.9$ | $60.0 \pm 27.9$ | 23.0 |

Table A12: **Visual Planning on iTHOR.** We report the mean success rate across 9 object navigation tasks in iTHOR. Each table entry shows the average success rate aggregated from 5 seeds. "$*$" denotes seen tasks during adaptation.

We provide additional experimentation of adaptation techniques on iTHOR (Eric et al., 2017), in which a mobile robotic agent is asked to perform egocentric navigation to a specified target object in different scenes. This benchmark poses challenges of navigating in partially observable settings and allows us to further evaluate adaptation methods on in-domain video generation from egocentric views. To perform adaptation, we reuse the video dataset provided by AVDC (Ko et al., 2024), which spans 12 target objects and includes 25 successful navigation trajectories for each object. We report the success rates of visual planning across 9 navigation tasks in Table A12, in which 4 tasks are unseen during adaptation. We provide a detailed list of iTHOR tasks along with their corresponding text prompts in Table A2. In Table A12, we again observe that the overall performance of both probabilistic adaptation and its inverse outperform that of in-domain-only baseline by a large margin, highlighting that the internet knowledge of pretrained video models can be effectively utilized for various downstream robotic applications through proper adaptation. This result further highlights how adaptation can be flexibly applied across varied robotic settings.

# I STEP COUNTS TO TASK SUCCESS IN CLOSE-LOOP VISUAL PLANNING

| Step Count w/ | Door Close* | Door Open | Window Close | Window Open | Drawer Close |
|---|---|---|---|---|---|
| In-Domain-Only | 80.0 | - | 176.0 | 344.0 | 25.3 |
| Vanilla AnimateDiff | 122.3 | - | 323.0 | 217.3 | 160.9 |
| Direct Finetuning | 96.0 | - | 159.2 | 333.3 | 63.8 |
| Subject Customization | 150.5 | - | - | 297.3 | 238.7 |
| Prob. Adaptation | 75.4 | - | 171.5 | 312.0 | 31.0 |
| Inverse Prob. Adaptation | 87.0 | - | 222.6 | - | 35.8 |
| Step Count w/ | Drawer Open | Coffee Push* | Soccer | Button Press | |
| In-Domain-Only | - | - | - | 204.0 | |
| Vanilla AnimateDiff | - | - | - | - | |
| Direct Finetuning | - | - | - | - | |
| Subject Customization | - | 155.0 | - | - | |
| Prob. Adaptation | 244.0 | 52.0 | - | 183.2 | |
| Inverse Prob. Adaptation | - | - | 144.0 | 192.0 | |

Table A13: **Step Counts of Visual Planning on MetaWorld.** We report the average number of taken steps in successful evaluation rollouts across 9 manipulation tasks in MetaWorld. Unsuccessful rollouts are omitted. We observed that probabilistic adaptation in general achieves task success using fewer number of steps.

