# OpenReview forum: "Solving New Tasks by Adapting Internet Video Knowledge"
_ICLR.cc/2025/Conference — ICLR 2025 Poster_

### Official Review · Reviewer_yUek · 2024-10-26

**Soundness:** 3
**Presentation:** 2
**Contribution:** 2
**Rating:** 6
**Confidence:** 3

**Summary:**

This paper explores a range of domain adaptation techniques and incorporates them into large-scale, pre-trained video models for domain-specific robotic tasks. Notably, the authors extend the motion module, originally introduced in AnimateDiff, to enhance the animation capabilities of the video model for these specific tasks. Building on a vanilla video model, the authors examine various domain adaptation strategies, including direct fine-tuning, subject-specific customization, and probabilistic adaptation. A key innovation in this work is the redesign of the probabilistic adaptation approach to an inverse version, which demonstrates superior performance across both visual planning and policy learning tasks. The inverse approach yields significant improvements, achieving a success rate of 52.6% in policy learning and 28.4% in video planning, along with enhanced visual quality in the MetaWorld environment.

**Strengths:**

1. This work explores an efficient strategy for adapting internet video knowledge to in-domain simulation environments.
2. The experimental design is thorough, encompassing evaluations of policy learning and video planning across both seen and unseen tasks.
3. Part of the experimental results look convincing, effectively demonstrating the performance of the proposed adaptation strategies.

**Weaknesses:**

1. The results in Table 1 show that subject customization achieves significant improvement over both the direct fine-tuning approach and vanilla AnimateDiff. However, probabilistic adaptation underperforms compared to the other three settings, exhibiting a notable discrepancy in return values. Although the authors assert that probabilistic adaptation focuses on in-domain estimation and struggles to learn effective policies in the Humanoid and Dog environments, I am concerned that the evaluation protocol may not be optimal when using policy discrimination for this setting. To my knowledge, probabilistic adaptation adopts score composition from both the pre-trained model and the domain-specific model, introducing a distribution shift during sampling. I recommend that the authors tune hyper-parameters for probabilistic adaptation methods by re-evaluating policy discrimination to minimize model prediction biases. Additionally, it would be helpful to see a case study that further examines the return values for probabilistic adaptation/inverse probabilistic adaptation, similar to the results presented in the Policy Discrimination section of the project website.


2. The results in Table 2 show that the inverse probabilistic adaptation model outperforms other adaptation methods in policy learning success rate on MetaWorld. Could the authors also provide the average returns for each baseline method?

3. Evaluation of Video Quality: The quantitative results in Table 3 are not entirely convincing, as only two evaluation tasks (coffee push and button press) are considered, which may lead to evaluation bias. Please provide additional video quality analysis across all tasks to present a more comprehensive assessment of the overall results.

4. Based on the qualitative results from both the project website and the paper, it does not appear that the in-domain-only model performs worse than probabilistic adaptation or inverse probabilistic adaptation in task-level motion; the primary differences seem to be at the pixel-level generation. Another concern is that the reproduced results of the in-domain-only model exhibit lower video quality compared to the qualitative results presented in the AVDC paper (synthesized videos in MetaWorld). This discrepancy could lead to an unfair comparison. Could the authors explain their implementation process in detail, noting any deviations from the AVDC paper that might account for this discrepancy in video quality? A thorough review of the reproduction process would help ensure an accurate assessment of the model's performance.

**Questions:**

While the authors provide a comparative analysis with baseline methods, the paper still lacks some thorough examination of adaptation techniques to demonstrate the significant improvement of video adaption to in-domain video generation and robot manipulation tasks. Additionally, some baseline methods were not accurately reproduced during the evaluation, raising concerns about the validity of the results.

Given these issues, I am inclined to give the rating below the acceptance threshold. However, I will reassess my rating after reviewing feedback from other reviewers and the authors' responses. I am willing to raise my score if the authors address these concerns.

---

> ### Author Response · Authors · 2024-11-21
> **Response to Reviewer yUek**
>
> We thank Reviewer yUek for their careful review and constructive feedback. We would like to address their concerns and questions below:
>
> **On Probabilistic Adaptation performance**: We update Table 1 with the addition of two new columns: results for inverse probabilistic adaptation, as well as utilizing the small in-domain model only.  Also, all table results now include mean and standard deviation computed from 5 seeds.  We have also carefully performed additional hyperparameter sweeps, including testing different values of prior strength as well as hyperparameters of Video-TADPoLe, for probabilistic adaptation, as the reviewer suggested; we still observe that the final policy supervision performance collapses.  However, as an approach, we discover that inverted probabilistic adaptation achieves nontrivial non-collapsing performance, and clearly outperforms just using the in-domain model.  Furthermore, its performance on Dog substantially improves over using vanilla AnimateDiff.  This suggests that this form of adaptation of score composition is still promising and applicable for Humanoid and Dog environments, and is not limited to MetaWorld.
>
> Furthermore, as the reviewer suggested, we provide policy discrimination visualizations of Probabilistic Adaptation and Inverse Probabilistic Adaptation for the best hyperparameter settings found on our updated [website](https://sites.google.com/view/videoadapt-iclr25), in the Policy Discrimination section.  We show that even then, probabilistic adaptation is weaker in distinguishing between expert and suboptimal trajectories both in terms of the magnitude of the difference between rewards calculated using expert and poor videos, across different frames. On the other hand, inverse probabilistic adaptation is much more adept at distinguishing between the trajectories, thus supporting its performance on the Dog environment when applied for policy supervision.
>
> **On Baseline Performance**: We update Table 2 to include in-domain-only baseline performance.  We verify that inverse probabilistic adaptation indeed still improves over utilizing only the in-domain model directly, particularly on unseen tasks such as window-open and window-close.  This direct comparison supports our hypothesis that adapting a cheap in-domain model trained on small-scale data with large-scale pretrained video models can help generalization to novel, unseen tasks.
>
> **On FVD Score**: In the original submission, two tasks were selected just to represent performance on tasks seen and unseen during adaptation, and help provide insight into their updated visual performance.  At the reviewers request, we have updated Table 3 to be computed across all 7 seen tasks (for the seen column) and all 7 unseen tasks (for the unseen column).  1000 synthesized videos were utilized to evaluate each (adapted) video model, for each task set (seen/unseen) for a more holistic picture of the trends.
>
> **On In-Domain Model Performance**: With the addition of the In-Domain Model only results for Table 1 and 2, we demonstrate that across both policy supervision and visual planning, adaptation is more favorable to using the in-domain model alone.  Noticeably, in Table 1 we observe that using in-domain alone noticeably underperforms compared to using Inverse Probabilistic Adaptation, and in Table 2 it underperforms both Probabilistic Adaptation and Inverse Probabilistic Adaptation.  We also highlight that policy supervision as an approach does not require strong pixel-level generation, in contrast to visual planning.

---

> ### Author Response · Authors · 2024-11-21
> **Response to Reviewer yUek (cont.)**
>
> **On AVDC reproducibility**: We reuse the open-source AVDC codebase, with the **exact same training hyperparameters and architecture configurations** that are used to generate the videos shown in that work.  We therefore do not believe that the codebase or implementation constitutes substantial discrepancies that may lead to unfair comparisons with alternative adaptation works.  However, the reviewer indeed notes that visual quality is slightly worse than when modeling RGB images directly.  We identify two main differences that can potentially cause this effect:
> 1. **Latent modeling**: Rather than modeling RGB images directly as in the original AVDC paper, we now use the in-domain model to model *latents* of images because we are now performing score composition with a *latent* video diffusion model (AnimateDiff).  Because each latent is a rich compression of RGB visuals, each latent bit affects the generation of multiple pixels when decoding back to pixel space.  Therefore, whereas slight inaccuracies of modeling RGB pixels (as in the original AVDC paper) may not be as noticeable to the naked human eye, such slight inaccuracies of modeling the latents can result in more noticeable artifacts of the final projected RGB image here.
> 2. **Latent dimensionality**: The RGB images AVDC was trained on for MetaWorld were of size [128x128x3] (Table 7 of the AVDC paper).  However, the latents utilized by AnimateDiff are of size (64x64x4) dimension.  Whereas technically each input image from the original AVDC contains 3 times as much data as each latent, this mismatch in input dimensionality may be a source of modeling discrepancy.
>
> **On examination of improvement**: As mentioned above, we supply baseline in-domain method performance, and demonstrate that adaptation techniques largely have improvement.  We have also clarified on our implementation of baseline methods, which reuses existing open-sourced code and settings, where the only change is what we model as input.  In our update draft, we highlight our thorough examination of adaptation techniques for facilitating robot manipulation tasks: we apply them across multiple environments and robot settings (e.g. DeepMind Control, MetaWorld, and iTHOR) as well as task types (continuous control, robotic manipulation, and navigation) and video viewpoints (3rd person, egocentric), evaluate their ability to facilitate text-conditioned generalization to novel tasks unseen during adaptation, and also examine suboptimal data usage, where we show how adapting large-scale video models to in-domain can significantly improve performance for downstream robotic tasks without explicitly requiring expert demonstrations. We also provide evaluations with an additional pretrained video model.  Across all these settings, we find that adaptation improves quantitative robotic downstream performance over using baseline approaches such as using in-domain video models or generally-pretrained text-to-video models alone.

---

> > ### Comment · Reviewer_yUek · 2024-11-24
> >
> > I appreciate the authors' efforts in addressing my concerns and providing supplementary results. Based on the responses and the results presented, I believe the proposed adaptation technique demonstrates promising generalization improvement for novel robot manipulation tasks. Therefore, I am inclined to raise my original rating to marginally above acceptance.
> >
> > However, the responses also raise some minor concerns:
> >
> > While the authors clarified that they employed the original AVDC settings and aligned the input latents with AnimateDiff for score composition, the training configuration of AVDC must adapt to the change in input dimensions (from pixel space to latent space). Relying on the original training configuration for the latent-based AVDC may degrade the quality of video modeling.
> >
> > In the revised Table 4, Vanilla AnimateDiff outperforms Direct Finetuning in both the overall success rate and the "Door Close" task (a seen task) in visual planning evaluation. This result appears counterintuitive, as Direct Finetuning is conducted in the in-domain video environment and should theoretically perform more accurately on seen tasks compared to the vanilla model.

---

> > > ### Author Response · Authors · 2024-11-25
> > > **Response to Reviewer yUek**
> > >
> > > We thank Reviewer yUek for their response, and we are happy to hear that we have addressed many of the listed concerns.  We appreciate the reviewer raising their score.  We seek to offer further clarifications on the additional listed points:
> > >
> > > **On latent-based AVDC**: in our preliminary experiments, we tried building latent-based AVDC off of two default AVDC in-domain model designs: the MetaWorld architecture and the iTHOR architecture (Table 7 of [1]).  The rationale for using the MetaWorld setting was that despite the input-dimension differences, the content the model was expected to handle was similar (the latents preserve spatial locality).  On the other hand, the motivation for building off of the iTHOR implementation was because it was designed to handle 64x64x3 RGB inputs and we were now seeking to model 64x64x4 latent inputs.  Ultimately we found that the resulting models were comparable in video modeling, and decided to continue using the MetaWorld setting for all further experiments.  However, we agree with the reviewer that further improvements for in-domain video modeling quality can be potentially achieved by tuning the in-domain model settings (architecture and hyperparameters) to fit latents more accurately.
> > >
> > > **On Direct Finetuning performance**: We indeed notice that while direct finetuning brings improvement over several tasks, the performance on “Door Close” alone decreases greatly, directly causing an overall average performance drop.  When investigating the generated visual plans, we found that direct finetuning is prone to suffering from overfitting; it is able to generate plans for initialization positions of the Door seen during pretraining, but struggles to generate coherent plans for the randomly initialized positions of the Door during visual planning.  On the other hand, for unseen tasks, the model potentially avoids this issue by relying on large-scale priors for generalization, as there are no provided demonstrations to copy.  Whereas this may potentially be mitigated by using early stopping or through further tweaking of finetuning parameters, we believe that avoiding overfitting is a serious consideration tied to the Direct Finetuning approach that is not as severe in other adaptation approaches.
> > >
> > > [1] Ko et al. Learning to Act from Actionless Videos through Dense Correspondences. ICLR 2024.

---

> > > ### Author Response · Authors · 2024-12-02
> > >
> > > Dear reviewer yUek,
> > >
> > > We really appreciate your constructive feedback to our submission and your continued engagement during the discussion period! As the discussion phase is coming to an end, we would likely to quickly follow up with you in case there are remaining questions we could help clarify, thank you!
> > >
> > > Best,
> > >
> > > The Authors

---

### Official Review · Reviewer_fa6P · 2024-11-03

**Soundness:** 3
**Presentation:** 4
**Contribution:** 3
**Rating:** 6
**Confidence:** 4

**Summary:**

This paper investigates three adaptation techniques for generative video models, and proposes two evaluation metrics that go beyond traditional visual similarity scores in an attempt to more precisely test the effect of different adaptation techniques of video models on downstream performance for robotic tasks. The proposed adaptation techniques require small amounts of training data, and in some cases (subject customization) only static paired data. The proposed evaluation methods are by using the adapted video model as either a visual planner or policy supervisor, each with its own strengths as noted at the end of the methods section. Experiments are performed on a dog/humanoid and robot arm dataset 2 tasks and 16 tasks respectively. The authors finally observe that domain specific fine-tuning (for the dog/humanoid) and inverse probabilistic adaptation (for the robot arm) achieve the best performance.

Overall this paper explores an important topic that will have many implications when using generative video models for robotic tasks.

**Strengths:**

1. Several robot arm tasks in MetaWorld environment and an additional environment are proposed as an evaluation scheme
2. The work is well motivated and tackles a nuanced but important question that is often glossed over in other policy-focused works.
3. Implementation details are very clearly stated and discussed so readers have a full picture
4. The subject customization technique is unique and novel, and I think the inverse probabilistic adaptation method is original (but please clarify this too (ask mentioned in Question #1 below)
5. Very interesting observations regarding visual appearance vs task success rate are made, motivating the need to perform real world experiments such as those done in this paper to really understand what the right way to train/fine-tune/adapt video models for robotic tasks.

**Weaknesses:**

1. Quantitative results are not presented for the out-of-domain Humanoid and Dog environments. Table 1 seems to be for in-domain dataset, and Figure 3 shows qualitative results, but quantitative results for the jump task would be good to include in Table 1.

2. Two different adaptation techniques are concluded to work the best for the two different environments. (fine tuning for the dog/human environment, and inverse probabilistic adaptation for the robotic manipulation environment). This raises 2 questions: first, why do the ‘best methods’ differ in these two environments? And second, what are readers supposed to take away when working with a new different environment? A discussion, about knowing what technique to use based on the environment and why, would be good to include.

3. Why do many adaptation methods completely fail for several tasks when using Video Planning (Table 4)? Especially the Soccer task? Is this due to poor inverse dynamics model or the inability to perform good video adaptation? There should be an explanation of why this is happening, otherwise it's hard to be convinced this is a valid experiment since the point of failure is unclear.

4. Subject customization is interesting, but it is unclear if motion dynamics in a new environment can be learned. The training data only includes static images paired with text, so when the dynamics vary greatly, will this method fail in comparison to direct fine-tuning/probabilistic adaptation? A good test of this would be to change the dynamics in an out-of-domain environment (something like the gravity constant if you can control that in the dog/humanoid world) and compare the 3 adaptation methods. I expect that static transfer cannot capture dynamics differences. If this is a tough experiment to run but there is a different way to prove the soundness of this adaptation method given different dynamics, please go ahead and present that, this was just one possible idea.

**Questions:**

Questions that should be addressed:
1. I am unsure if (inverse) probabilistic adaptation has been explored before at all in the robotics tasks context. Maybe make this more explicit in the ‘adaptation techniques for diffusion models’ related works section.
2. Eq 1 & 2 introduces notation but many of the variables (theta, t, tao) are not defined. Please state what each variable means.
3. Also, please clarify what e_theta is in Eq 1&2. Is this indicating the denoising unet for the AnimatedDiff model?
4. It would be good to clarify what ‘out of domain’ means in these adaptation experiments. It seems the robot arm and its dynamics must stay fixed for subject customization since the visual token is encoding this specific robot arm right? What about for (inverse) probabilistic adaptation, are there any such constraints on the domain you are adapting to?
5. Sec 3.2 does not have details about how the reward is calculated using the adapted video model, all details are in Appendix B. Maybe it is because VideoTADPoLe is unknown to me, but I expect this to be the case for many readers. So please give atleast a brief description in the main paper and then you can point the readers to Appendix B.
6. Please include the exact details of the ‘small dataset’ mentioned in line 312. I believe is important and should not just be kept in supplemental because the paper focuses on how to adapt with small # of examples.
7. Please show a qualitative example from the Subject Customization for Figure 3 so that the readers  can see the qualitative comparisons as described in the paragraphs in Sec 4.2.
8. What is ‘high variance’ referring to in L285? Variance in the output of a video model planner?
9. AnimateDiff produces very short horizon videos (in terms of time into the future). Therefore, completing a task seems to require lots of closed-loop rollouts. A mention of the number of steps to complete the task might be good to include in each type of experiment.

Minor Comments:
1. It is unclear whether AnimateDiff is the right model for this work. Why did the authors not choose to just use a video-model itself such as StableDiffusionVideo which has stronger priors about long-term motions? Some mention of this would be good to include.
2. The ‘Studying Data Quality’ section is interesting, but very few details are mentioned in the main paper, with most left to the supplementary section. If you would like to keep this section in the main paper, I would suggest at least including what a ‘suboptimal dataset’ means. Otherwise there is very little a reader can take away from this section.

---

> ### Author Response · Authors · 2024-11-21
> **Response to Reviewer fa6P**
>
> We thank Reviewer fa6P for their careful review, and for their thoughtful questions and suggestions for strengthening our work.  We provide clarifications and additional additional results below:
>
> **Quantitatively Evaluating Novel Text-Conditioned Behaviors**: we would like to clarify that in contrast to walking, a ground-truth reward function to evaluate jumping is not provided by the environment.  In fact, a main motivation for our work is if leveraging some known demonstrations from the environment (such as walking) can facilitate generalization to novel tasks with no predefined ground-truth reward functions through adaptation (such as jumping).  On the other hand, for an environment where we can directly test text-conditioned generalization in a quantitative way, we therefore heavily utilize MetaWorld, where there are a variety of task settings **with ground-truth success evaluation**.  Here, we quantitatively compare the ability of adaptation techniques to generalize to tasks unseen during adaptation in a text-conditioned manner, both through visual planning and policy supervision approaches.  Furthermore, in our updated experiments (Appendix G, Table A10) we include an **additional task** that enables us to quantitatively evaluate task generalization performance for novel tasks unseen during adaptation - navigation in the iTHOR environment.  We also present such results in our general response, verifying the efficacy of adapting large-scale pretrained video models for generalizable robotic behavior.
>
> **On the “best” method, and main takeaways**: In Table 1 we have updated the columns to show mean and std calculated over 5 seeds, and we have since included an additional column, for inverse probabilistic adaptation results.  We discover that our proposed inverse probabilistic adaptation achieves reasonable performance on the Dog and Humanoid environments, with significant improvement over using Vanilla AnimateDiff in the Dog experiment.  Furthermore, it achieves the best overall policy supervision performance for MetaWorld, and competitive performance for visual planning in MetaWorld, as well as the best performance when using suboptimal data.  We believe that as a takeaway, inverse probabilistic adaptation is a safe choice to use for adaptation across multiple environments.  However, beyond the “best” overall, we also position our work as a study on the expected adaptation performance given considerations on adaptation data and resources (e.g. when direct finetuning is computationally infeasible, or when only suboptimal demonstrations are accessible).  We discover that surprisingly, when only still frames of the subject are available, subject customization can still achieve nontrivial improvements for both dog/humanoid and robotic manipulation settings.  As subject customization only adapts to the visual characteristics of the environment, this result may be because the environment dynamics of dog/humanoid are similar to those of natural videos, enabling the reuse of large-scale motion priors trained from internet-scale video datasets without modification.
>
> **Explaining visual task planning failures**: For failed tasks, after inspecting the executed interactions against the visual plans, we found that they still follow reasonably closely - suggesting that task failure does not stem from an inaccurate inverse dynamics model.  Furthermore, for probabilistic adaptation, the planned motions themselves also look reasonably respectful of in-domain dynamics.  Rather, in many failed tasks, the video plans simply do not appear to perform the specified task (e.g. the robot arm appears to reach towards the net rather than go low towards the soccer ball - failure case visualizations have been included in our updated [website](https://sites.google.com/view/videoadapt-iclr25) at the very bottom).  This suggests that rather than the inverse dynamics, or the in-domain motion modeling, the bottleneck may be the learned text-motion alignment in the utilized backbone video model, and perhaps additional improvements in downstream robotic performance can directly follow from further advancements in the field of text-to-video modeling.

---

> ### Author Response · Authors · 2024-11-21
> **Response to Reviewer fa6P (cont.)**
>
> **On subject customization for novel dynamics**: we follow the reviewer’s suggestion to investigate subject customization when the environment dynamics have been modified, by decreasing the gravity constant in the Humanoid environment.  We observe that subject customization is still able to learn to maintain the humanoid to be upright and maintains reasonably high ground-truth reward; however, its walking performance is not as high as under normal gravity conditions.  We provide such visualizations, and aggregate quantitative results over 5 seeds on our updated [website](https://sites.google.com/view/videoadapt-iclr25), where it still seems to take step-like motions, but does not make much distance progress most likely due to the change in environment dynamics.  We can also imagine that in the extreme, when gravity does not exist at all, or when gravity is overwhelmingly strong, walking is impossible at all.  We believe this result is because for subject customization, the model has simply updated based on the environment's visual characteristics, but preserves the motion module from large-scale pretraining.  Such large-scale motion priors may be generally applicable across environments, but may not be as robust if the environment exhibits severe out-of-distribution dynamics from its training data (such as floating).  This may explain why subject customization performs well in the regular humanoid/dog environments, but for robotic settings like MetaWorld that may require further understanding of dynamics different from what was seen during pretraining of the large-scale video model, in-domain dynamics supervision approaches such as probabilistic adaptation and its inverse perform much better.
>
> **Paper Clarifications**: we have made numerous edits to our draft in concordance with the reviewer’s thoughtful suggestions.  We update the related work to make clear that inverse probabilistic adaptation is a method we propose, we introduce notation and clarifications for the listed Equations in Section 3.1.1.  We also clarify that in Equations 1 and 2, $\epsilon_{\theta}(\cdot)$ denotes the trainable small in-domain model and $\epsilon_{\text{pretrained}}(\cdot)$ denotes the frozen, pretrained video model (AnimateDiff in our work) in section 3.1.3. We also provide more description regarding Video-TADPoLe in Section 3.2.2 and directly reference Appendix B for further details. For direct finetuning, we utilize the same number of in-domain videos for probabilistic adaptation and its inverse **to form the small dataset**, and we clarify the detailed number of videos used for specific tasks in Section 4.1 “Benchmark” paragraph.
>
> **Clarifying “out-of-domain”**: We denote all tasks as ‘out of domain’ if no demonstrations or examples of such tasks were provided during the adaptation procedure.  For example, in MetaWorld, we provide examples (whether just still frames, or demonstrations) of a certain set of tasks (denoted with asterisks in Table A2) and then apply them to unseen tasks (denoted without asterisks in Table A2).  Noticeably, certain tasks involve novel objects or table setups unseen during adaptation such as Window or Drawer, where performance may benefit from adaptation with large-scale video models that have observed similar objects during pretraining.
>
> **On Dynamics and Constraints**: Subject customization leverages a large-scale pretrained motion prior that is applied to arbitrary environments with arbitrary dynamics; the video model has only been adapted to the specific visual characteristics of the environment (but not its dynamics).  Therefore, the dynamics do not have to stay fixed, as any dynamics setting is agnostic to the large-scaled pretrained video model’s motion module.  However, we can observe that supplying in-domain dynamics information is useful, evidenced by the performance of probabilistic adaptation and its inverse on MetaWorld.   Furthermore, we demonstrate that any dynamics information alone, without requiring them to be expert demonstrations, can already adapt large-scale pretrained video models to become better facilitators for robotic behavior - this is shown through the suboptimal data experiments, where demonstrations largely do not even solve the tasks at all, but plausible dynamics of the robotic arm is communicated.  We show how inverse probabilistic adaptation is not largely hindered by such constraints, being able to be applied meaningfully to continuous control (humanoid/dog), robotic manipulation, and the additional navigation tasks (iTHOR), and without assuming expert demonstrations (suboptimal experiments).

---

> ### Author Response · Authors · 2024-11-21
> **Response to Reviewer fa6P (cont.)**
>
> **On Qualitative Examples of Subject Customization for Dog Jumping**: We update an additional freeform generation achieved by subject customization in the “Novel Text-Conditioned Generalization” section on our [website](https://sites.google.com/view/videoadapt-iclr25), in which the video again reflects the natural “jumping” motion as specified by the text prompt.
>
> **On "High Variance"**: here, high variance refers to the policy optimization process.  Estimating the gradient estimate for the policy parameters depends on trajectories (and therefore rewards) achieved through stochasticity, such as sampling actions from the policy over many timesteps (and potentially stochastic transition dynamics or reward functions in certain environments).  We highlight that although using video models to supervise the learning of policies does not necessarily require high-quality in-domain visual generation ability, by virtue of performing policy learning it may suffer from the known issue of high variance gradient estimates.  On the other hand, visual planning does not learn a policy and avoids this pitfall, but benefits from high-quality in-domain video generation ability to generate coherent plans.
>
> **Avg Steps to Complete Task**: We update our Appendix to include the average number of action steps to complete the task, for all successful executions, in Table A11.  Unsuccessful executions are avoided in our aggregates, as they merely time-out.  As the reviewer suggests, reporting the number of steps not only expresses the expensiveness of applying adapted video models as visual planners, but also provides a sense of the “expertness” of the video plan.  We discover that, of successful trajectories, probabilistic adaptation generally has shorter execution plans.
>
> **On AnimateDiff vs. StableVideoDiffusion**: We select AnimateDiff for our experiments because it is a powerful, text-conditioned video-generative model that has demonstrated strong zero-shot text-conditioned generalization capabilities.  We identify text-conditioning as a way to specify the robotic task/behavior of interest - for example, describing “a robot arm opening a door” versus “a robot arm closing a door”.  On the other hand, StableVideoDiffusion has strong motion priors, but the open-sourced version does not support text-conditioning.  We are therefore unable to use it to explicitly specify tasks of interest, whether for visual planning or policy supervision, for testing task generalization capabilities after adaptation.  We update our work to include using a different text-conditioned video model, AnimateLCM, for visual planning on MetaWorld and report the results in Table A9 of our updated Appendix. We discover that AnimateLCM performance is further improved by inverse probabilistic adaptation, with even better performance than that achieved when using AnimateDiff with inverse probabilistic adaptation.  This highlights the general benefits of adaptation techniques, untied to particular backbone video diffusion models.
>
> **On Details of Suboptimal Dataset**: we have added a brief description of suboptimal dataset generation in the main body, for the readers clarity on our setup in the main text, and kept our link to Appendix A for further details.

---

> ### Author Response · Authors · 2024-11-25
> **Follow-Up**
>
> Dear Reviewer fa6P,
>
> We gently invite you to review our additional experiments, updated draft, and points of clarification, as your feedback is invaluable for strengthening our work.  We hope that we have sufficiently addressed your concerns, and if there remain any further questions we would be more than happy to provide additional explanations.
>
> We appreciate your time and consideration,
>
> The Authors

---

> > ### Author Response · Authors · 2024-12-02
> >
> > Dear Reviewer fa6P,
> >
> > We would like to offer another gentle nudge to review our updated experimental results, revised draft, and clarifying responses, which we believe addresses your previous listed concerns. With tomorrow being the last day for reviewer responses, if there are any additional questions or feedback that we can help address, we would greatly appreciate your engagement.
> >
> > Thank you!
> >
> > The Authors

---

### Official Review · Reviewer_NuKB · 2024-11-03

**Soundness:** 2
**Presentation:** 3
**Contribution:** 2
**Rating:** 6
**Confidence:** 3

**Summary:**

The paper presents a study on how to incorporate pre-trained, language conditioned video generation models into robot learning. The paper considers three adaptation strategies for using video generation towards policy learning: finetuning the majority of the video generation model on expert demonstrations (most expensive for data and compute), subject customization using a few static images from the target domain, and probabilistic adaptation which trains a small in-domain model supervised by the large-scale video generation model. The paper further considers two options for downstream robotic task evaluation: video planning by predicted a plan to follow into the future and policy supervision by using the adapted video model to synthesize rewards for the policy. The different adaptation and evaluation techniques are benchmarked on the MetaWorld-v2 and DeepMind Control Suite simulations. Probabilistic adaptation and subject customization show promise in this domain when combined with the AnimatedDiff video generation model.

**Strengths:**

* The general problem of incorporating large-scale video generation into robot policy learning is of high interest to the robotics community. Particularly, with the rise in popularity of World Models in robotics, this paper's study is quite timely.
* The paper is well written and fairly easy to follow.
* I found the discussion of positives and negatives of the policy supervision evaluation approach to be insightful in Sec. 3.2.2.
* The experiments are fairly extensive in the simulated environments and transparent regarding underperformance in certain domains.
* The limitation section is thoughtfully constructed.

**Weaknesses:**

* The majority of the assumption in the paper seems to be that video models provide strong motion priors and just require adaptation to the downstream domain visually (please correct me if I'm wrong) e.g., through static images in subject customization. However, from my experience large-scale video generation models still really struggle with the dynamics of the environment. This even seems to come across in the MetaWorld-v2 benchmark where a lot of the tasks achieve 0% or very low success rate.
* All the results are based on a single video generation model: AnimateDiff. It would have been more compelling to show results on a few models, in case the findings are particular to the specific chosen model.
* Some of the language is quite repetitive throughout the paper (e.g., last sentence of 3.1.1 and first sentence of 3.1.2).
* I am not sure I entirely agree with the premise that it is impossible/highly challenging to obtain in-domain video demonstrations for a task. ~100 demonstrations are often assumed for imitation learning papers like Diffusion Policy [1], particularly if high success rates are desired on dexterous tasks. This relates to the low success numbers seen in the MetaWorld-v2 tables, particularly for tasks unseen during adaptation. Optimization compute cost seems like potentially a bigger concern.
* Since a large focus of the paper is evaluating video generation adaptation in robotics tasks, it feels like real-world robot experiments should be included in the analysis. In simulation, adaptation may be more focused on the appearence of the simulated environment than the physics modeling of the language-specified task. In the real-world, modeling the physics of the interactions between the robot and the environment in real-time is the challenge.
* The data budgets for the expert demonstrations is quite low and appears rather arbitrary (25 for MetaWorld-v2 and 6 and 17 for the DeepMind Control Suite). It would be helpful to ground these design choices in recent literature.
* Standard error is only present for the first column of Table 1. It should be included across the considered methods.
* The very poor performance of probabilistic adaptation in Table 1 is attributed to the small capacity of the in-domain model used. However, this should have been explored to understand the potential of this method in the DeepMind domain.
* The results in Figure 3 were not particularly convincing to me for the dog jumping case. It seems the video generation model was able to generate a partially reasonable but not entirely accurate jumping dog sequence. The resulting simulated behavior appears to be a degenerated version of that suboptimal supervision. Although I do see that there is potential there for simulating unseen behaviors zero-shot.
* Showcasing results where all methods get 0% (or close to 0^%) success rate for about half the tasks is not very meaningful as a takeaway. It seems like these experiments should have been iterated on futher. Likely including additional expert demonstrations would help the success rate.

Some typos and points of confusion are listed below:
1. Line 018 - video modeling ... [encodes].
2. The score in Sec. 3.1.3 should be formally defined.
3. The FVD acronym should be defined once in line 239.

[1] Chi, Cheng, et al. "Diffusion policy: Visuomotor policy learning via action diffusion." The International Journal of Robotics Research, 2023.

**Questions:**

1. Is there a reason batch-balanced co-training was not studied as an alternative adaptation technique? Co-training has seen decent success in robot learning in recent years [2]?
2. Why is inverse probabilistic adaptation good fro MetaWorld-v2 but so much worse for the DeepMind Control Suite?

[2] Khazatsky, Alexander, et al. "DROID: A large-scale in-the-wild robot manipulation dataset." RSS, 2024.

---

> ### Author Response · Authors · 2024-11-21
> **Response to Reviewer NuKB**
>
> We thank Reviewer NuKB for their thorough, thoughtful comments, questions, and suggestions.  We seek to address the outstanding concerns:
>
> **On the fundamental assumption**: we would like to clarify that we do not explicitly make this assumption; rather, in our paper we seek to investigate to what extent this is the case.  Therefore, beyond just adapting with visual characteristics alone (subject customization), we also study techniques that explicitly model in-domain motions (such as probabilistic adaptation and its inverse).  A takeaway is that indeed supervising in-domain motions can give superior performance (e.g. inverse probabilistic adaptation performance in Table 2), but a surprising additional takeaway is that just adapting to visual information such as subject customization alone can have substantial improvement across multiple environments and robotic settings (Table 1 and Table 2).  As the large-scale pretrained motion prior is unchanged via subject customization, this highlights the strength of leveraging large-scale pretrained video models for downstream robotic applications, and how benefits can be achieved with cheap adaptation requirements (e.g. just static images).  Furthermore, while we demonstrate that adapting video models to in-domain motions can improve downstream performance, we also study to what degree expert motions are necessary or if arbitrary dynamics-informing demonstrations are sufficient.  In our Suboptimal Data experiments (End of Section 4.3, Appendix A), we verify the surprising result that any in-domain dynamics information, not necessarily just expert ones, is useful for adapting large-scale pretrained video models for robotic tasks.  Therefore, we believe that **both** in-domain dynamics and visual information are useful for adapting video models for generalizable downstream robotic task performance.
>
> **On additional video models**: We agree that ablating over video models is an interesting direction.  We update our submission to include additional experimental results for visual planning using a different video diffusion model - AnimateLCM [1].  In Table A9, we discover that both Probabilistic Adaptation and its Inverse achieve improved performance over just leveraging vanilla AnimateLCM, with Inverse Probabilistic Adaptation performing the best.  This suggests that our adaptation explorations are not tied to a particular choice of large-scale text-to-video model backbone.
>
> **On writing**: we thank the reviewer for highlighting how our writing can be improved; we have made adjustments to the draft accordingly (e.g. in the example mentioned, we have removed the redundant last sentence of 3.1.1 entirely).  We have also incorporated the typos listed further down in the reviewer’s comment. We have now introduced the denoising/score function used in Section 3.1.3 in Section 3.1.1 with consistent notation.  Edits to the draft are shown in blue font.
>
> **On the availability of in-domain video demonstrations**: we agree that for settings where demonstrations can be easily achieved, imitation learning such as Diffusion Policy can be explicitly applied.  However, there may exist settings where generating in-domain video demonstrations is difficult - such as explicitly controlling the DeepMind Control Suite Dog to jump.  Moreso, rather than just difficulty as a consideration alone, imitation learning requires the collection of explicit demonstrations for each new task of interest; even if each individual task is not difficult to collect data for, this can quickly become intractable if the **set of tasks of interest** is large.  We therefore are interested in how large-scale pretraining, such as leveraging large-scale pretrained video models trained on internet-scale data, can facilitate generalization to novel tasks without explicitly seeing examples of such tasks (in which case imitation learning can be directly applied).  Rather than just imitating provided demonstrations, we seek to leverage them (via adaptation) to also generalize to novel task behaviors that were not explicitly demonstrated; this is more tractable and feasible than collecting new data for imitation learning for every new task of interest.  Furthermore, we agree that optimization and compute cost are of interest in this adaptation; we therefore highlight the performance achieved by each adaptation technique while considering the training data and resource requirements they need.  We believe our study across these considerations is useful to the resource-constrained practitioner.
>
> [1] Wang et al. AnimateLCM: Computation-Efficient Personalized Style Video Generation without Personalized Video Data. arXiv:2402.00769 2024.

---

> ### Author Response · Authors · 2024-11-21
> **Response to Reviewer NuKB (cont.)**
>
> **On Real World Experimentation**: we agree that real-world robot experiments are interesting, and believe our method has direct implications for actual robotic experiments.  At the same time, however, we would like to highlight that our insights and contributions are still meaningful within these simulated robotic settings.  We demonstrate that even in simulation, beyond just adapting to visual characteristics, modeling the movement and interaction physics helps for enabling downstream robotic performance (such as via inverse probabilistic adaptation).  Furthermore, while we agree that the interaction dynamics between the robot and environment will be key in adaptation for real robotic experiments, we still expect to adapt to the appearance of the robot arm and its environment, as we do not assume such data is present during large-scale pretraining of the backbone text-to-video model.
>
> **On data budget**: we agree that the design decisions should be more standardized, and have updated our results in the paper to reflect this.  In our updated results, visualized in Table 1, we have standardized the demonstrations for continuous control (e.g. Humanoid and Dog) to be 20 for each environment, across all adaptation methods (e.g. 20 still frames for Subject Customization).  Prior literature has not proposed a standardized number of demonstrations for the Humanoid and Dog tasks, to our knowledge, but 20 falls within a reasonable range for per-task demonstrations.  We are also interested in studying adaptation with a low-data budget (in the extreme case, Subject Customization only uses still-images); requiring fewer in-domain demonstrations lowers the barrier of utilizing general, large-scale pretrained text-to-video models for downstream robotic applications across environments and behaviors of interest.
>
> **On Standard Error**: in our updated draft, we update the entries of Table 1 to include standard error calculated across 5 seeds for consistency.
>
> **On poor in-domain model for humanoid and dog**: we have updated Table 1 to include policy supervision performance when using in-domain only and inverse probabilistic adaptation results on the Humanoid and Dog environments.  We therefore empirically verify that in-domain performance achieves poor performance, suggesting its inability to capture the breadth of the complex dynamics within its limited capacity.  Both the Humanoid and Dog environments are well-known to have complex transition dynamics in comparison to simpler environments such as Hopper/Walker/Cheetah, or the fixed robot arm in MetaWorld.  Furthermore, our updated inverse probabilistic adaptation results demonstrate non-collapsed performance, and clearly outperforms using the in-domain model only.  Its performance on Dog also substantially improves over using vanilla AnimateDiff.  This suggests that this form of adaptation of performing score composition is still promising and applicable for Humanoid and Dog environments, and is not limited to MetaWorld.
>
> **On suboptimal dog jumping behavior in Figure 3**: we agree that the video model itself only appears to generate a partially reasonable sequence for a dog jumping.  We hypothesize this stems from the fundamental capabilities of the large-scale text-to-video model; and that adaptation has still demonstrated benefits in translating the beliefs of a large-scale text-conditioned video model into something the agent can achieve within the environments domain (e.g. the agent still does seem to be following the video model’s belief after adaptation, even if the video model’s belief is imperfect).  We therefore believe that with advancements in large-scale text-to-video models, both in their natural motion priors as well as their ability to align them with text specifications, the accurate completion of downstream robotic tasks should directly improve as well via adaptation.
>
> **On tasks with zero success rates**: we include such tasks for complete transparency on the limitations of our investigated adaptation approaches; if we were to remove them and only show successful tasks, it may mislead the reader into believing all MetaWorld tasks could be solved through some form of adaptation.  We agree that the performance for such experiments could be improved with access to in-domain demonstrations - and if lots of such demonstrations are assumed, then other techniques such as imitation learning could be applied.  However, our main goal is to study the performance that can be achieved on novel tasks without associated prior demonstrations but through the adaptation of large-scale generally-pretrained text-to-video models.  We also investigate if expert demonstrations are necessary and find that surprisingly, large-scale motion priors and text-conditioning capabilities from powerful text-to-video models can still be adapted with suboptimal in-domain data (End of Section 4.3, Appendix A, Table A1) to facilitate successful performance on novel robotic tasks.

---

> ### Author Response · Authors · 2024-11-21
> **Response to Reviewer NuKB (cont.)**
>
> **On co-training**: we thank the reviewer for pointing us to the co-training technique. In the DROID [2] paper, co-training is used to improve the policy robustness in “out-of-distribution” tasks, by mixing the training batch data with both *in-domain* task demonstrations and a *large-scale* dataset collected using a specific robot platform in a balanced manner. In our setup, we treat the videos (or images) used for adaptation as in-domain data, however, we **do not** assume the availability of a large-scale dataset. Instead, we utilize a large-scale pretrained video model for adaptation, which can be interpreted as a summarization of the large-scale dataset it is pretrained on and with powerful capabilities of generating novel samples. This provides us with a more flexible way of leveraging the knowledge stored in the large-scale data for task generalization. On the other hand, when only a large-scale dataset is accessible, co-training can be applied.
>
> **On Inverse Probabilistic Adaptation Performance**: as discussed above, we have since updated Table 1 to include inverse probabilistic adaptation performance for the DeepMind Control Suite, with results averaged over 5 seeds, and found that it is able to outperform using the in-domain model alone.  It also achieves a substantial performance improvement on Dog over using vanilla AnimateDiff.
>
> [2] Khazatsky, Alexander, et al. "DROID: A large-scale in-the-wild robot manipulation dataset." RSS, 2024.

---

> ### Author Response · Authors · 2024-11-25
> **Follow-Up**
>
> Dear Reviewer NuKB,
>
> We gently invite you to review our additional experiments, updated draft, and points of clarification, as your feedback is invaluable for strengthening our work.  We hope that we have sufficiently addressed your concerns, and if there remain any further questions we would be more than happy to provide additional explanations.
>
> We appreciate your time and consideration,
>
> The Authors

---

> > ### Comment · Reviewer_NuKB · 2024-11-25
> > **Response to Rebuttal**
> >
> > Thank you to the authors for the effort to put in additional experiments, run several seeds, and report standard errors on the results. These efforts have improved the scientific rigor of the paper.
> >
> > My main concern is the inconsistency of the best performing method across the different experiments, leaving me confused on the takeaway from the paper. Furthermore, although I do appreciate the transparency of the difficulty of, for example, the MetaWorld tasks, I worry that these tasks have not been given a fair try to succeed. Is there existing work that also reports 0% success rate on these tasks or is the setting considered in this paper significantly more difficult than prior work? If the latter, is such a difficult setting able to convey a meaningful message to the community?
> >
> > Thank you once again for all the hard work put in; it does show. However, prior to considering changing my score, I would want to see a clear, consistent insight that a reader would be able to takeaway from the paper.

---

> > > ### Author Response · Authors · 2024-11-27
> > > **Response to Reviewer NuKB**
> > >
> > > We deeply appreciate your continued constructive feedback on our work!
> > >
> > > **On the main takeaways:**
> > >
> > > Our work offers two main takeaways messages to the readers: First, when in-domain video demonstrations are available, inverse probabilistic adaptation serves as a robust approach to leverage “Internet video knowledge”. It consistently outperforms the in-domain only baseline across all environments, different video generative models (AnimateDiff and AnimateLCM), for both policy learning and visual planning, and with either expert or suboptimal demonstrations. Second, when only in-domain image examples are available, subject-customization serves as a surprisingly strong baseline, especially when policy learning is used.
> > >
> > > As the reviewer noticed, when zooming into the detailed performance breakdowns, there are additional interesting trends (or “outliers”), such as: (1) Subject customization is especially desirable when the in-domain model is weak (Table 1); (2) Probabilistic adaptation may sometimes outperform its inverse version (Table 2), although both outperform the in-domain only or Internet only baselines; (3) Subject customization appears to be more effective for policy learning than visual planning, the latter of which intuitively requires adaptation of in-domain “dynamics”. We believe our thorough and candid study is on its own a contribution, and we will clarify our writing to make all contributions digestible to the readers.
> > >
> > > **On the 0% success rate tasks on Meta-World:**
> > >
> > > We largely follow the setup from TADPoLe (our policy learning framework) and use the same set of tasks for Meta-World. In Table 2 and A12 (newly added, with one setting having a positive success rate across all tasks) of our manuscript, we can see that although a certain baseline algorithm achieves 0% success rate for some tasks, each of the tasks we report has at least one algorithm with non-zero success rate. We believe including all tasks would help readers better appreciate the pros and cons of individual algorithms. To put the results in context, in the AVDC work, Table 1, the authors reported performance of Behavior Cloning, UniPi, and Diffusion Policy, where each baseline approach has at least one, and up to four tasks with zero success rate. Their setup is easier than ours, in the sense that *each task* has 15 video demonstrations.
> > >
> > > Additionally, we agree with the reviewer that Meta-World is indeed challenging under our setup, hence we add iTHOR evaluations to better support our findings.

---

> > > ### Author Response · Authors · 2024-12-02
> > >
> > > Dear Reviewer NuKB,
> > >
> > > We hope that our previous response clarifying our MetaWorld findings and summarizing our main insights was able to adequately address your concerns, and we hope it factors positively in your score adjustment considerations. With tomorrow being the last day for reviewer responses, if there remain any additional questions or feedback that we can help address, we would greatly appreciate further engagement!
> > >
> > > Thank you!
> > >
> > > The Authors

---

> > > > ### Author Response · Authors · 2024-12-03
> > > > **Thank you!**
> > > >
> > > > Dear Reviewer NuKB,
> > > >
> > > > We just noticed that you updated the rating to borderline accept, thank you!
> > > >
> > > > We understood that the discussion period has ended. Regardless, please feel free to let us know if you have other feedback to our manuscript (e.g. in the final official review), and we will make sure to incorporate them (along with what we already provided in the rebuttal and the revised manuscript) in the final version.
> > > >
> > > > Best,
> > > >
> > > > The Authors

---

### Official Review · Reviewer_DMT5 · 2024-11-05

**Soundness:** 3
**Presentation:** 3
**Contribution:** 2
**Rating:** 5
**Confidence:** 3

**Summary:**

This work investigates how to adapt large-scale, pre-trained video models to solve novel, text-conditioned robotic tasks in specific environments. The authors explore three adaptation techniques: direct ft, subject customization, and probabilistic adaptation. It further introduces inverse probabilistic adaptation. The model is evaluated on metaworld and deepmind control suite.

**Strengths:**

1) The motivation is straightforward.
2) The paper is well written, it does lots of controlled experiments to discuss which adaption techniques are better.

**Weaknesses:**

The scope of this paper is limited. While visual planning and goal-conditioned imitation/reinforcement learning (IL/RL) are indeed significant topics in robotics, the approach to generating visual targets appears straightforward.

The experimental results lack novelty, as it is unsurprising that a web-scale, pre-trained text-to-image (T2I) or video generator would produce superior images or videos.

Furthermore, this work does not directly address whether improvements in image quality contribute to enhanced performance or generalization in control tasks. For example, while Table 2 shows that inverse probability achieves the highest success rate, its FVD score in Table 3 is higher than others, raising questions about the relationship between quality and control success. Additionally, the study does not include generalization tests for control tasks. Although the paper aims to address new tasks, no genuinely novel tasks are explored.

**Questions:**

My suggestion is to conduct more experiments, such as testing model generalization ability.

---

> ### Author Response · Authors · 2024-11-21
> **Response to Reviewer DMT5**
>
> We thank Reviewer DMT5 for the helpful comments and suggestions.  We update our manuscript with updated details, and also seek to assuage the concerns of the reviewer in our response.
>
> **On paper scope**: we consider the general scenario of two learning paradigms: visual planning, where an actionable plan, which could be argued as a *dynamically updated* visual target, is generated; and policy supervision, where the video model is used as a discriminative reward provider without explicitly synthesizing any visual targets at all.  The approaches we investigate to adapt the Internet video knowledge are versatile and general, and can be applied in both learning paradigms. Finally, we would like to refer the reviewer to the ICML 2024 position paper Video as the New Language for Real-World Decision Making (Yang et al.) [1] where they outlined multiple challenges when applying video generation as a tool for decision making. Our work makes concrete contributions to address the “dataset limitations” and “limited generalization” abilities of applying video generation for decision-making (Section 5 of [1]).  We do so by only leveraging small amounts of in-domain data for adaptation, and utilizing powerful large-scale pretrained video models to demonstrate how adaptation can be successfully across a variety of robotic environments and tasks (continuous control, robotic manipulation, and navigation).
>
> **On experimental surprise**: we respectfully disagree. Our goal is not to “produce superior images or videos” for arbitrary domains, as suggested by the reviewer. Our goal is to adapt video generative models for particular environments and embodied agents, both of which are highly unlikely to be covered by web-scale pre-training data. It is thus not obvious if and how pre-trained video priors would help. In practice, we observe that: (1) Pre-trained video models failed to generate coherent videos conditioned on initial environmental state; (2) Models trained on in-domain videos that are multiple orders of magnitude smaller can outperform web-scale pre-trained models on solving tasks through visual planning; (3) Different adaptation techniques have significant impact on the task success rates. All of these signals suggest that it is neither obvious that web-scale models will always help visual planning or policy learning, nor it is clear how to properly integrate web video priors.
>
>
> **On the relationship between generation quality and task success**: we would like to highlight that we do not assume FVD scores as an effective indicator for task success; we therefore propose using downstream performance on robotic tasks as another important metric to evaluate and understand adapted video generative models beyond simply looking at visual quality metrics as was done in prior works [2].  In fact, by comparing Table 3 and Table 4 in our updated draft, we can observe that in visual planning there is not always a positive correlation between FVD scores and task performance, and good visual quality alone is not sufficient to achieve task success. How well the generated video plan can properly reflect the motion described by the text prompt is equally or even more critical for task solving.  Furthermore, beyond visual planning, we investigate how adapted video models can act as policy supervisors that provide zero-shot text-conditioned rewards to inform in-domain policy learning. In this setting, we only utilize video models to perform one-step score prediction in the latent space for reward computation rather than any explicit generation in RGB space.  As mentioned in Section 3.2.2, video models do not necessarily need the ability to create high-quality in-domain videos from scratch to behave as effective policy supervisors; they simply need to be able to critique the quality of achieved in-domain frames.  Thus, expressing adapted knowledge through rewards may allow the detachment of downstream policy performance from demands on video generation quality.
>
> [1] Yang et al. Video as the New Language for Real-World Decision Making, Position Paper, ICML 2024.
>
> [2] Yang et al. Probabilistic Adaptation of Text-to-Video Models. ICLR, 2024.

---

> ### Author Response · Authors · 2024-11-21
> **Response to Reviewer DMT5 (cont.)**
>
> **On generalization tests for control tasks**: In our evaluations (e.g. Table 2, Table 4 and Figure 3), we have provided quantitative and qualitative results for tasks that were **unseen** during the adaptation process. We consider these tasks to be **novel** to the pre-trained video models even after adaptation, and evaluations on these tasks remain valid generalization tests for adapted video models. For example, in Figure 3 we have shown a video generated by an adapted video model that faithfully reflects the novel behavior described by the text prompt “a dog jumping”, along with a successful policy rollout. These results have demonstrated that the tested adaptation methods are able to effectively incorporate in-domain information into pretrained video models while maintaining their internet-scale knowledge and generalization capabilities to achieve strong performance on these novel tasks.
>
> **On additional experimentation**:  In Appendix G of our updated draft, we provide additional experiments on 9 object navigation tasks in the iTHOR environment [3], in which a mobile robotic agent is asked to perform egocentric navigation to specified target objects across various scenes. We also present the experimental results regarding navigation success rates in our general response. Both probabilistic adaptation and its inverse outperform in-domain-only baseline by a large margin, which again showcases the effectiveness of adaptation in leveraging web-scale knowledge to facilitate in-domain tasks. We also highlight egocentric navigation as a new task class to test adaptation over, distinct from the previously reported continuous control and robotic manipulation settings, showcasing the flexibility of adaptation across a variety of robotic task types.
>
> [3]  Kolve et al. AI2-THOR: An Interactive 3D Environment for Visual AI. arXiv:1712.05474, 2017.

---

> ### Author Response · Authors · 2024-11-25
> **Follow-Up**
>
> Dear Reviewer DMT5,
>
> We gently invite you to review our additional experiments, updated draft, and points of clarification, as your feedback is invaluable for strengthening our work. We hope that we have sufficiently addressed your concerns, and if there remain any further questions we would be more than happy to provide additional explanations.
>
> We appreciate your time and consideration,
>
> The Authors

---

> > ### Author Response · Authors · 2024-12-02
> >
> > Dear reviewer DMT5,
> >
> > As the discussion phase is coming to an end, we would like to offer another gentle nudge to consider our responses to your review, along with our revised manuscript with updated results and new evaluations on iTHOR. Thank you.
> >
> > Best,
> >
> > The authors

---

> > > ### Comment · Reviewer_DMT5 · 2024-12-03
> > > **Response to rebuttal**
> > >
> > > Thank you for your careful rebuttal. I see some of my concerns have been addressed by the rebuttal, however, I think the experimental setting, especially for the metaworld simulator, is way too easy that weakens the contribution of this work. Therefore I raise my score to 5.

---

> > > > ### Author Response · Authors · 2024-12-03
> > > > **Thank you and a quick follow-up**
> > > >
> > > > Dear Reviewer DMT5,
> > > >
> > > > Thank you so much for engaging with us and for upgrading your rating!
> > > >
> > > > We would like to quickly clarify that we believe our evaluation setup, as pointed out by reviewers NuKB and fa6P, is very challenging, especially on Meta-World. We actually responded to the reviewers explaining the challenges (of task generalization), and why our results convey meaningful insights that are broadly interesting to the community, despite the challenge of our evaluation setup.
> > > >
> > > > We welcome the reviewer to briefly check our discussions with reviewer NuKB (key word "On the 0% success rate tasks on Meta-World") and fa6P (key word "Explaining visual task planning failures) to learn more, thank you!
> > > >
> > > > (For what it is worth, we will post a follow-up message that provides more details on the challenges, thank you for bearing with us!)
> > > >
> > > > Best,
> > > >
> > > > The Authors

---

> ### Author Response · Authors · 2024-12-03
> **More details, in case helpful**
>
> We would like to provide further clarifications on **comprehensive** and **challenging** nature of our experiment setting:
>
> **On the evaluated tasks and benchmarks**: We select a *wide range* of tasks and benchmarks, spanning continuous locomotion (on Dog and Humanoid), robotic manipulation (on MetaWorld) and egocentric navigation (on iTHOR), to comprehensively evaluate adaptation techniques for downstream task performance. These benchmarks, including MetaWorld, are widely adopted by previous works in a similar context, and are recognized as meaningful testbeds under equal or less challenging setups compared to ours. For example, AVDC [1] reports visual planning results on MetaWorld having trained an in-domain model across **all** evaluated tasks, whereas in our work we evaluate a more challenging setup; we evaluate on tasks for which we do not provide demonstration data.  Furthermore, we provide experimentation for settings where only suboptimal demonstration data is provided, whereas prior works all assume expert demonstrations.  We hope the reviewer can appreciate the *breadth of our experimentation*, as well as the *more challenging nature of our experimental setup* compared to prior work in this area.
>
> **On the challenges posed in our setups**:  We would like to highlight that the challenges in our setups are exhibited across different levels. Firstly, the web-scale data that the large video models are pretrained on can be drastically different from the downstream domains during evaluation, and we hope to bridge this gap by adapting very limited domain-specific examples. Secondly, the selected evaluation benchmarks pose complexity and difficulties from different aspects, achieving competitive performance across all benchmarks is challenging. Thirdly, prior work (e.g. [1]) selects both MetaWorld and iTHOR benchmarks as we do. However, unlike their setup which trains and tests on the same set of tasks, we include considerable evaluations performed on tasks that are unseen during adaptation, to further showcase the generalization capabilities. Especially, in MetaWorld experiments (Table 2, Table 4, Table A9, Table A12), we provide results on 7 out of 9 tasks (denoted with no asterisk) that the model has never observed before the evaluation. Furthermore, we perform adaptation with suboptimal in-domain demonstrations but require the adapted video model to act optimally during evaluation (Table A1).
>
> To summarize, our experimental design is largely built upon those adopted by prior works, but poses more challenges to fully demonstrate the effectiveness and generalization capabilities of adaptation techniques in solving domain-specific robotic tasks.
>
> [1] Ko et al. Learning to Act from Actionless Videos through Dense Correspondences. ICLR 2024.

---

### Author Response · Authors · 2024-11-21
**General Response**

We thank all reviewers for their constructive feedback - we are glad that our work was recognized as “well-motivated” and “insightful”, and our experimental design and execution were recognized as “extensive” and “thorough”.

We have since updated the paper to reflect the comments raised by the reviewers, notably for notation clarification and dataset details.  We also evaluate using an additional backbone text-to-video model, AnimateLCM [1], which we report in Appendix F, Table A9.  We discover that inverse probabilistic adaptation achieves the best overall performance and outperforms both using an in-domain model and the large-scale AnimateLCM model by themselves, reconfirming the efficacy of adaptation in improving in-domain task performance.  This further demonstrates that adaptation as an approach can be applied flexibly across different backbone text-to-video models for successful downstream robotic applications.

Furthermore, we include new experimental results for a novel environment and task setting - navigation in iTHOR [2].  We note that the navigation task is a distinct challenge from the previously investigated approaches of continuous locomotion (DMC) and robotic manipulation (MetaWorld), as the videos are now egocentric in nature and successful task performance relies on handling partial observability.  In Appendix G, Table A10, we show that across 9 object navigation tasks, the overall performance of both probabilistic adaptation and its inverse outperform that of in-domain-only baseline by a large margin, highlighting that the internet knowledge of pretrained video models can be effectively utilized for various downstream robotic applications through proper adaptation.  We provide the table in Markdown format below, in which unseen tasks are denoted in italic format:

| Success Rate (%) w/  	| Spatula  in Kitchen  | Toaster  in Kitchen | Painting  in Living Room | Blinds  in Bedroom | ToiletPaper  in Bathroom | *Pillow  in Living Room* | *DeskLamp in Living* Room | *Mirror  in Bedroom* | *Laptop  in Bedroom* | Overall |
|--------------------------|:--------------------:|:-------------------:|:------------------------:|:------------------:|:------------------------:|:----------------------:|:-----------------------:|:------------------:|:------------------:|:-------:|
| In-Domain-Only       	|  	13.3 ± 29.8 	| 	33.3 ± 33.3 	|         	0        	| 	13.3 ± 29.8	|    	40.0 ± 36.5   	|   	6.67 ± 14.9  	|   	6.67 ± 14.9   	|      	0     	| 	26.7 ± 27.9	|   15.5  |
| Prob. Adaptation     	| 	20.0 ± 29.8  	| 	60.0 ± 27.9 	|         	0        	| 	26.7 ± 36.5	|   	73.3 ± 14.9    	|   	13.3 ± 18.3  	|   	13.3 ± 18.3   	|      	0     	| 	53.3 ± 29.8	|   **28.9**  |
| Inverse Prob. Adaptation |  	13.3 ± 18.3 	| 	33.3 ± 23.6 	|         	0        	| 	33.3 ± 33.3	|    	40.0 ± 14.9   	|   	6.67 ± 14.9  	|   	13.3 ± 18.3   	| 	6.67 ± 14.9	| 	60.0 ± 27.9	|   23.0  |


We thank the reviewers for their time and consideration, and welcome further discussion,

The Authors





[1] Wang et al. AnimateLCM: Computation-Efficient Personalized Style Video Generation without Personalized Video Data. arXiv:2402.00769 2024.

[2]  Kolve et al. AI2-THOR: An Interactive 3D Environment for Visual AI. arXiv:1712.05474, 2017.

---

### Author Response · Authors · 2024-11-25
**General Response - Additional Experimental Results**

We have updated the draft with additional experimentation results.  In Appendix I, Table A12, we include quantitative policy supervision results on MetaWorld through adaptation with another video model backbone, AnimateLCM.  This complements our previous experiment of utilizing adapted AnimateLCM for visual planning (Appendix F, Table A9).  We discover that despite the change in text-to-video model backbone, inverse probabilistic adaptation consistently achieves the best success-rate performance and relative performance trends across methods are maintained.  With AnimateLCM, inverse probabilistic adaptation obtains the highest overall success rate of 65.2%, surpassing all other evaluated video models and adaptation techniques, with non-zero success rates across **all** evaluated tasks.

We invite the reviewers to engage with our updated draft and results, as well as our clarification responses, and look forward to their additional feedback and discussions regarding our work.

The Authors

---

### Meta-Review · Area_Chair_vgbD · 2024-12-18

**Metareview:**

The paper investigates techniques for adapting large-scale internet video models to facilitate novel robotic behaviors through text conditioning, earning a recommendation for acceptance from all reviewers after discussion. The work demonstrates how video models pretrained on internet-scale data can be effectively adapted using limited in-domain examples to enable text-conditioned generalization for robotic tasks, evaluated through both visual planning and policy supervision approaches.

The reviewers consistently praised the comprehensive experimental evaluation across different environments (DeepMind Control, MetaWorld, iTHOR), the thorough analysis of different adaptation techniques, and the paper's strong motivation given the growing importance of video models in robotics. Initial concerns focused on the inconsistency of best-performing methods across environments, lack of clarity around implementation details, and questions about baseline comparisons.

The authors provided extensive responses and additional experiments that satisfied most reviewer concerns. They added evaluations with a new video model (AnimateLCM), expanded results on iTHOR navigation tasks, clarified implementation details particularly around latent-space modeling, and demonstrated more comprehensive FVD evaluations across all tasks. The authors also effectively explained the challenging nature of their experimental setup compared to prior work, particularly for MetaWorld tasks.

While some minor concerns remain about training configurations for latent-space modeling and direct fine-tuning performance on seen tasks, the overall consensus is that this work makes a valuable contribution by thoroughly evaluating how internet-scale video knowledge can be adapted for robotic tasks.

**Additional Comments On Reviewer Discussion:**

None -- see metareview

---

### Decision · Program_Chairs · 2025-01-22

Accept (Poster)